# Identification of molecular nociceptors in *Octopus vulgaris* through functional characterisation in *Caenorhabditis elegans*

Eleonora Maria Pieroni[1,2], Vincent O'Connor[1], Lindy Holden-Dye[1], Pamela Imperadore[3], Graziano Fiorito[3] and James Dillon[1,*]

## ABSTRACT

Nociception, a phenomenon crucial for animal survival, deploys evolutionarily conserved molecular mechanisms. Among invertebrate species, cephalopods are of particular interest as they possess a well-developed brain speculated to be able to encode pain-like states. This has led to their inclusion in the Directive 2010/63 EU for welfare protection.

However, the molecular mechanisms of nociception in cephalopods are still poorly characterised and it is important to address this knowledge gap to better understand cephalopods' capacity to express pain states. Here we describe a bioinformatic strategy utilising conserved nociceptive genes, to identify the orthologous candidates in the *Octopus vulgaris* transcriptome. We identified 51 genes we predict to function in nociception. These add to the mechanosensory TRPN and the unique chemotactile receptors recently identified in octopus suckers, thus expanding the set of genes that merit further functional characterisation in cephalopods. We therefore selected 38 orthologues in *Caenorhabditis elegans*, a tractable experimental platform and tested loss of function mutant strains of distinct functional gene classes (e.g. *osm-9, egl-3, frpr-3*) in a low pH avoidance paradigm. This identified 19 nociceptive-related genes to be prioritised for further functional characterisation in *O. vulgaris*.

KEY WORDS: Nociception, Sensory cues, Octopus, *O. vulgaris*, *C. elegans*, Model

## INTRODUCTION

The ability to detect potential noxious stimuli in the environment involves the triggering of specialised sensory neurons called nociceptors, which activate a simple reflex response that organises a withdrawal behaviour of the animal from a potential threat (Sherrington, 1906; Dubin and Patapoutian, 2010). The number and type of modalities of noxious stimuli detected by these cells is encoded by the molecular determinants, which define the transduction and integration of such environmental cues (Wood,

2006; Dubin and Patapoutian, 2010). The crucial adaptive value of nociception in contributing to increase the individual survival, made it a conserved feature across Eumetazoa (Smith and Lewin, 2009).

In cephalopods, evidence of nociception is found in *ex vivo* and *in vivo* experimental observations in which noxious cues delivered to the arms or mantle of the animal are able to trigger withdrawal responses but also associated neurophysiological features such as post-injury sensitization (Crook et al., 2013, 2014; Alupay et al., 2014; Howard et al., 2020). As an example, acetic acid exposure in *ex vivo* preparations of octopus cut arms is able to trigger a reflexive retreat of the limb (Hague et al., 2013). Similarly, a complex response is suggested to be organised in the central brain to produce secondary protective behaviours such as grooming and concealing of the arm after acetic acid exposure in alive octopuses, which suggests a discriminative component of pain (Crook, 2021). The increasing interest in nociception and its processing in these cephalopods derives from the recognition of their sophisticated neuroanatomical central organization. This likely entails a top-down regulation of modulatory nerve signalling underlying the potential to exhibit pain pathways. Due to this possibility, as a precautionary principle, cephalopods are currently subject to legislation that protects their welfare when used in research (European Parliament and Council of the European Union, 2010; Smith et al., 2013; Birch et al., 2021). However, to fully address the question around pain perception, a broader understanding of the molecular determinants of nociception in cephalopods is required. To date, only two molecular classes of sensory detectors have been experimentally identified in *Octopus bimaculoides* (van Giesen et al., 2020) and *Sepioloidea lineolata* (Kang et al., 2023). These are located in the sensory epithelium surrounding the suckers, which harbours putative cellular nociceptors that morphologically resemble mammalian sensory neurons (Rossi and Graziadei, 1956). One was identified as a NompC (TRPN) orthologue, activated by mechanical stimulation whilst the other class was found to be a phylum-specific ion channel, sensitive to both attractants and aversive cues (Kang et al., 2023). This provides evidence of both conserved and exclusive molecular components of sensory detection in cephalopods (van Giesen et al., 2020).

Most of the direct investigation of nociceptive processing in cephalopods is constrained by their limited experimental genetic tractability. These limitations derive from a series of challenges posed by culturing these animals, especially considering the very delicate early life stages (Iglesias et al., 2007; Vidal et al., 2014) and the strict requirements this species needs in terms of accommodation, nutrients, and temperature supplies (Ponte et al., 2022). Therefore, currently most of the work carried out on *Octopus vulgaris* is based on animals that are caught from the wild in a non-standardised way (Pieroni et al., 2022; Sykes et al., 2023). This generates two major issues; one is the complete absence of control over their genetic background and the second one is represented by

[1]Faculty of Environmental and Life Sciences, School of Biological Sciences, University of Southampton, Southampton SO17 1BJ, UK. [2]Association for Cephalopod Research 'CephRes' ETS, Via Rampe Brancaccio 49, 80132 Napoli, Italy. [3]Department of Biology and Evolution of Marine Organisms, Stazione Zoologica Anton Dohrn, Via Francesco Caracciolo, 80121 Napoli, Italy.

*Author for correspondence ( jcd@soton.ac.uk)

E.M.P., 0000-0002-2258-2780; V.O., 0000-0003-3185-5709; L.H.-D., 0000-0002-9704-1217; G.F., 0000-0003-2926-9479; J.D., 0000-0003-3244-7483

the variable animal welfare status after capture and transport. However, the highly conserved genetic basis of nociception across the animal phyla provides an opportunity to take an indirect, but nonetheless, informative approach.

Here we conducted an *in silico* analysis to identify molecular candidates for nociception in the *O. vulgaris* transcriptome. To provide insight into their function we identified their *Caenorhabditis elegans* orthologues and tested loss of function mutants for deficits in a simple assay of chemical nociception by exposing mutant nematode strains to different noxious cues, including low pH, a common aversive cue in both species (Sambongi et al., 2000; Hague et al., 2013; Crook, 2021).

The approach using low pH highlighted 19 nociceptive-related genes prioritised for functional characterisation, reinforcing the molecular conservation of gene families in distinct organisms and mapping out a platform through which experimentally intractable cephalopod genes can be investigated in *C. elegans*.

## RESULTS

### Resolution of a query set of conserved Eumetazoan nociceptive-related genes

Our bioinformatic strategy, identified a total number of 474 nociceptive related genes collectively retrieved from literature analysis (141), databases (200) and Gene Ontology (GO) search (133).

A considerable number of retrieved genes (152 entries) were found multiple times across the different sources or under synonyms. These duplications were removed, resulting in a total number of 322 distinct candidates (Fig. 1A and Table S1A).

We then filtered down this list by focusing on essential nociceptive and anti-nociceptive determinants, such as receptors directly gated by noxious stimuli or known modulators of sensory detection. This was done with a view of excluding categories of proteins involved in multiple physiological pathways or with a rather secondary/diffuse role in nociception. Among these, neurotransmitters, and their receptors (29 genes), transcription factors, large families of enzymes involved in widespread cellular activities (165 genes) and molecules contributing to the inflammatory responses were excluded (29 genes). Furthermore, given the challenge posed by some candidates in distinguishing between isoforms of the same gene or different representatives of the same subfamily we pooled related genes (e.g. TRPV1-6 have been pooled into TRPVs, ASIC1-3, into ASICs, PIEZO1-2 into PIEZO) for a total of 25 genes.

Altogether, this filtering produced a final number of 74 potential candidate genes that were blasted against *O. vulgaris* transcriptome (Fig. 1A and Table S1B).

### *O. vulgaris* shares most of the conserved protein families implicated in nociception

Each selected protein sequence from the query set was blasted against *O. vulgaris* transcriptome to find orthologue genes (Fig. 1B).

Out of 74, we found 53 orthologues with at least one representative in *O. vulgaris* (Table S1B). This corresponded to 177 identified transcripts in the *O. vulgaris* transcriptome assembly. We manually curated each of them to remove incomplete sequences (35) and found additional, related sequences (29), which brought the final number of refined complete transcripts to 171 (Table S2). The corresponding 51 distinct genes encoded for proteins belonging to more than 30 different families (and 50 subfamilies) of receptors and modulators associated with sensory detection of chemical, mechanical and/or thermal stimuli. The results were organised into

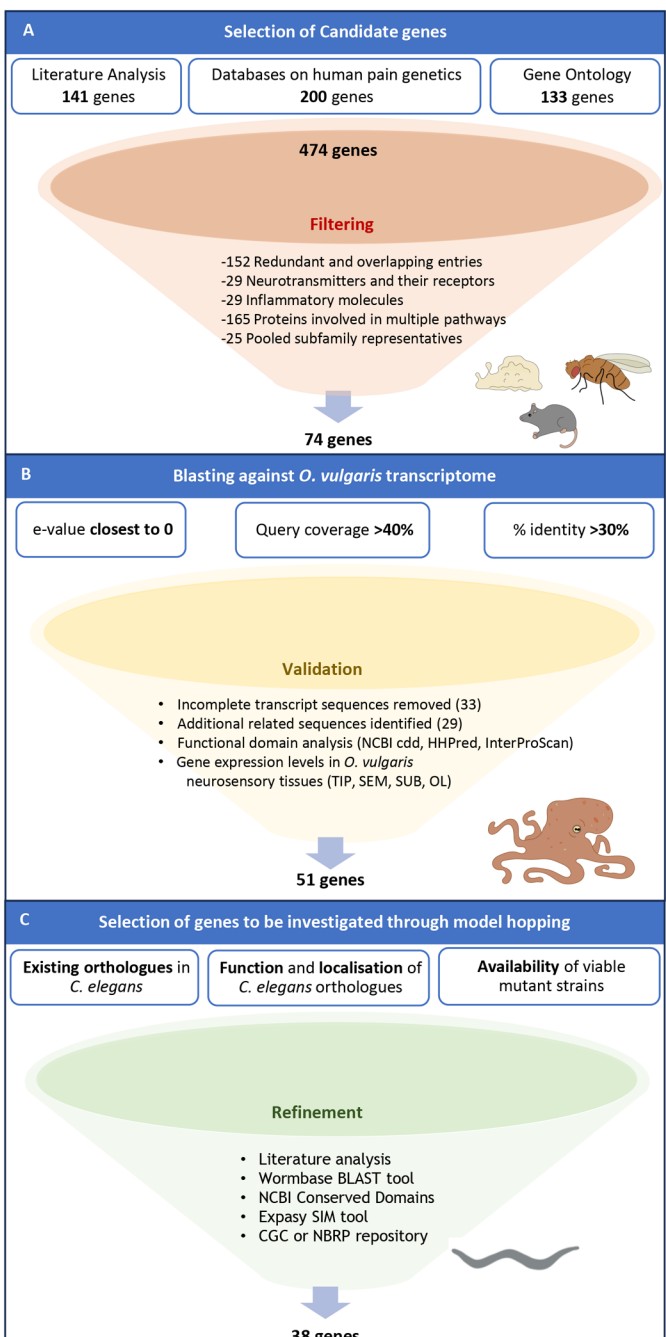

**Fig. 1. Summary of *in silico* analysis leading to the selection of *O. vulgaris* putative molecular nociceptive determinants to be modelled in *C. elegans*.** (A) Literature analysis of well-characterised Eumetazoan nociceptors was complemented with information from databases on human pain genetics (Human Pain Genetic Database, IASP Pain research, MOGILab) and GO "gene and gene products" to produce 474 distinct candidate genes. We manually filtered to exclude overlapping entries among resources (152), genes involved in multiple pathways (165), neurotransmitters and their receptors (29) and those involved in the inflammatory component of pain (29). We additionally pooled together (25) different representatives of the same subfamily (e.g. ASIC1-3, TRPV1-8 etc.). (B) The resulting 74 candidates were blasted against *O. vulgaris* transcriptome to retrieve the closest orthologues, which were then manually curated, leading to 51 candidate genes. We prioritised genes based on available tissue expression levels in octopus sensory and nervous tissues (TIP, SEM, SUB, OL – for details see Table S2). (C) To promote the functional characterisation in *C. elegans*, mutant strains for the orthologue genes were selected. This filtering led to a final number of 38 genes across 45 strains.

different categories based on the putative biological role of these proteins: 1) classical activators of nociception, 2) voltage-gated ion channels/subunits, 3) proteins related to neuropeptide and lipid metabolism, and 4) other modulators of nociception (Table S2).

Belonging to the first category, orthologues of well-characterised proteins such as transient receptor potential (TRPs) and amiloride sensitive (ASCs) ion channels were identified with nine and three representatives (within multiple isoforms), respectively.

The presence of specific subunits of ion channels implicated in nociception, such as the calcium voltage-gated channel subunit α2δ subunit 3 (CACNA2D3), known for its role in thermosensation was found, whilst other known channels specifically involved in nociception triggering such as $Na_v1.7$-$Na_v1.9$, were not clearly identified, mostly leading to $Na_v1.1$ (SCNA1) and $Ca_v1.1$ (CACNA1S) orthologues (Table S2).

### Genes implicated in canonical analgesic pathways are missing from *O. vulgaris* transcriptome

Out of the 74 selected genes, 21 did not have a single representative in *O. vulgaris* transcriptome. These include opioid precursors and derivatives, classical opioid receptors, and endocannabinoid receptors. However, receptors that share strongest homologies with those receptors, such as Allatostatin C receptor (AstCR) with an assigned 'opioid receptor' protein family membership (IPR001418) in InterProScan, and an Opioid growth factor receptor (OGFR) were identified. Furthermore, enzymes such as neprilysin (NEP), carboxypeptidase (CPE), neuroendocrine convertase (PCSKs) in the case of neuropeptides-related proteins or diacylglycerol lipase (DAGL) and N-acyl-phosphatidylethanolamine-hydrolysing phospholipase D (NAPE-PLD) in the case of lipid modulators were found. This shows the existence of a conserved pathway for the biosynthesis, processing, and metabolism of neuropeptides and lipids that have a broader physiological role, which may include nociception modulation and its physiological mitigation.

### *C. elegans* mutants of *O. vulgaris* orthologue genes were selected for characterisation in a chemical aversion assay

Following the identification and manual curation of *O. vulgaris* candidates, we looked at the data on available tissue distribution and prioritised the analysis of genes shown to be enriched in neurosensory tissues (tip of the arm, TIP; supra-oesophageal mass, SEM; sub-oesophageal mass, SUB; and optic lobe, OL), especially when more than one hit resulted in a plausible orthologue candidate of the putative nociceptive gene (Table S3).

As we intended to model the response triggered by these genes in *C. elegans*, each resulting *O. vulgaris* protein sequence was blasted against *C. elegans* genome database to find, where present (Table S3), the closest nematode orthologue (Fig. 1C). The result of this final refinement led to 38 *C. elegans* genes, for which we selected 45 different strains (Table 1).

Based on the recurring evidence showing nociceptive responses triggered in octopuses through acetic acid administration (*O. vulgaris*, Hague et al., 2013, *O. bocki*, Crook, 2021), we selected low pH (M9, pH 3) as the exemplar cue and systematically tested against selected *C. elegans* strains using the chemoaversion drop assay.

### The drop assay is a sensitive test to study chemoaversion in *C. elegans*

We analysed wild-type (WT) N2 and selected mutants in the assay and benchmarked the results against previously published data to validate our approach. The results agreed with published data (Sambongi et al., 2000) and showed that low pH elicited a significant aversive response in N2 (M9, pH3 average number of reversals=3.990±0.115) when compared to the control (M9, pH 7, average number of reversals=0.432±0.033). This allowed us to set the threshold for measuring responsiveness (1 for avoidance, 0 for defective avoidance) to at least three reversals for M9 pH3 (Fig. S1).

We then verified the sensitivity of the drop assay in discriminating the responsiveness of different strains of worms. We first tested the well-characterised *che-3 (e1124)*, a strain with ultrastructural defects in the ciliated dendritic endings of the amphid sensory neurons that play an essential role in chemosensation and compared its performance to the WT N2 (Perkins et al., 1986; Wicks et al., 2000).

As previously published for *che* mutant strains (Sambongi et al., 2000), *che-3 (e1124)* showed a poor avoidance performance to acidic pH when compared to the WT N2 (*P*<0.0001) with less than 30% responsiveness to low pH (M9, pH 3 average number of reversals 2.293±0.130), a difference of more than 50% with the WT N2 (around 80% responsiveness Fig. S1).

Additionally, we analysed *C. elegans* responsiveness to 200 mM Na acetate ($C_2H_3NaO_2$) to check which chemical component of acetic acid was responsible for triggering the aversive response. Worms showed no sensitivity to sodium acetate (200 mM Na acetate versus Ctrl M9 pH7, *P*=0.9802 in WT N2, *P*=0.9924 in *che-3 (e1124)*) demonstrating that the repulsive response of M9, pH3 was elicited by the protons ($H^+$) rather than by the acetate ion (Fig. S1).

### *C. elegans* is a discriminating platform for characterising conserved molecular nociceptors

Once we established the drop assay as a valuable test to discriminate the responsiveness of the strains, we analysed the performance of all the 45 strains of *C. elegans* mutants that came out of the list of 38 orthologues of *O. vulgaris* putative nociceptive genes (Fig. 2 and Fig. S2). The drop assay identified 19 distinct genes across the four different categories that seem implicated in the detection or processing of acidic pH.

Eight out of the 14 strains belong to the 'classical activators of nociception' category with impaired strains, all showing an average number of reversals below the set threshold for responsiveness (Fig. 3).

Mutant strains belonging to the TRP channel subfamilies, including the TRPVs *osm-9* (q= 0.0002) and *ocr-2* (q=0.0002), the TRPA *trpa-1* (q<0.0001) and the TRPMs *gon-2* (q=0.0002) and *gtl-2* (q=0.0002), all showed a significant impairment in the detection of acidic pH (Fig. 2A and Fig. S2A). Representatives of TRPN (*trp-4*), TRPC (*trp-1*) and TRPP (*pkd-2*) TRP subfamilies, however, did not show any defect in pH 3 aversion.

Worm strains of the orthologues of other families of classical pH detectors such as ASCs, did not show any defect in chemoaversion as it might be expected by acidic sensing ion channel representatives. In fact, *mec-4* and *mec-10* strains exhibited on average more than three reversals [3.2000 *mec-4 (u253)*, 3.1000 *mec-4 (e1611)*, 5.1000 *mec-10 (e1515)*, Fig. 3]. The only strain showing lack of response to pH 3, was the neuronal and glial ASC double mutant *acd-1 (bz90)/deg-1(u38u421)* (q=0.0002). Finally, *pezo-1 (av143)* was also found to be significantly impaired in the response (q= 0.0092).

An impaired response to pH3 was also found among specific strains representative of the 'voltage-gated ion channels/subunits' category. This included *kqt-3* (q=0.0002) and *shk-1* (q=0.0002) representatives of the KCNQ and KCNA sub families of potassium channels. This impact was selective as neither *kqt-2* (q=0.8707) nor

**Table 1. List of *C. elegans* loss of function mutant strains for the selected nociceptive related genes selected from *O. vulgaris* transcriptome after manual filtration of the *in silico* analysis**

| Gene name | Function | *O. vulgaris* candidate | *C. elegans* orthologue | Allele | Strain name | Mutation | Reported behavioural phenotype | Gene expression | % domain identity |
|---|---|---|---|---|---|---|---|---|---|
| **Classical activators of nociception** | | | | | | | | | |
| Acid Sensing Ion Channel (ASIC) | Ion channel involved in low pH detection (Deval and Lingueglia, 2015; Lee and Chen, 2018) | c28071_g1_i1 MN081801 | mec-4 | e1611 | CB1611 | Substitution (Ala442Thr). Causes aggregation of protein product (Driscoll and Chalfie, 1991) | Insensitivity to gentle touch; lower escape response to plate taps; altered NaCl chemotaxis (Chatzigeorgiou et al., 2010) | Anterior and posterior mechanosensory neurons (ALM, PLM, PVM, PVD) (Lai et al., 1996) | 28.3% identity in 127 residues overlap; Score: 126.0; Gap frequency: 2.4% |
| | | | | u253 | TU253 | Deletion (363 bp). Eliminates the first hydrophobic domain affecting the selectivity of the Na+ filter (third exon) (Hong et al., 2000) | Insensitivity to gentle touch; suppressed nose touch response; enhanced acuity to low dilutions of AWC–sensed odours; reduced response to low dilution of AWA sensed odours (Árnadóttir et al., 2011) | | |
| FMRFamide-activated sodium channel (FaNaC) | Ion channel activated by FMRFamide. Potentially involved in anti-nociceptive responses in invertebrates (Cottrell et al., 1990; Lingueglia et al., 1995) | c34210_g4_i4 MN081866 | mec-10 | e1515 | CB1515 | Substitution (Ser105Phe). missense in the DEG/ENaC domain - loss of function (Huang and Chalfie, 1994) | Severe light touch insensitivity; significant reduction in the mechanoreceptor currents (Árnadóttir et al., 2011) | Anterior and posterior mechanosensory neurons (ALM, PLM, PVM, PVD), (FLPL/R and PVDL/R) (Huang and Chalfie, 1994) | 37.7% identity in 53 residues overlap; Score: 105.0; Gap frequency: 0.0% |
| | | | acd-1/deg-1 | acd-1 (bz90) deg-1 (u38tu421) | - | bz90: Deletion (1625 bp) including the region of DEG/ENaC domain. u38: null mutation (Wang et al., 2008) | Individual loss of function deg- is moderately defective in low pH detection. However, double glial/neuronal mutant acd-1/deg-1 is dramatically unresponsive to acidic pH (Wang et al., 2008) | ASK, ASG, ADL, ASI (Wang et al., 2008) | 21.5% identity (46.4% similar) in 455 aa residues overlap |
| Transient Receptor Potential A (TRPA) | Ion channel involved in the activation by heat, cold, mechanical and chemical stimuli (Kwan et al., 2006) | c31382_g11_i1 MN081859 | trpa-1 | ok999 | RB1052 | Deletion (1334 bp). Eliminates the N terminal region (Kindt et al., 2007) | Defects in mechanosensory behaviours related to nose-touch responses and foraging (Kindt et al., 2007) | PVD, PDE, PHA, PHB, ASH, OLQ, IL1 (Kindt et al., 2007) | 33.6% identity in 837 residues overlap; Score: 871.0; Gap frequency: 8.4% |
| Transient Receptor Potential C (TRPC) | Ion channel involved in the activation of nociceptors (Julius, 2013; Ramsey et al., 2006; Sun et al., 2020) | c30608_g5_i7 PV164455 | trp-1 | sy690 | TQ225 | Deletion (>2 kb). Eliminates first 4 exons including promoter region and most of N terminus (Feng et al., 2006) | Reduction in nicotine-response (Feng et al., 2006). Unknown involvement in aversive sensory detection | Interneurons, motor neurons, pharyngeal neurons, sensory neurons, muscles (Xiao and Xu, 2009) | 51.9% identity in 770 residues overlap; Score: 1935.0; Gap frequency: 3.9% |
| Transient Receptor Potential M. (TRPM) | Ion channel responsive to cold (TRPM8), heat and some irritants (TRPM2 & TRPM3). (Julius, 2013; Ramsey et al., 2006; Tan and McNaughton, 2016) | c31340_g2_i1 PV164447 | gon-2 | q388 | EJ1158 | Substitution (Glu955Lys). Loss of function mutation Lambie et al., 2015 (Lambie et al., 2015) | Gonad defective (Sun and Lambie, 1997) Hypersensitivity to Mg++ (Teramoto et al., 2005) Unknown involvement in nociception | Pharynx, excretory cells, intestine (Xiao and Xu, 2009) | 31.2% identity in 186 residues overlap; Score: 233.0; Gap frequency: 8.1% |
| | | c31482_g3_i4 PV164444 | gtl-2 | tm1463 | LH202 | Deletion (463 bp). Eliminates from 5th to 7th exon (Teramoto et al., 2010) | Hypersensitivity to Mg++ (Teramoto et al., 2005) Unknown involvement in nociception | Pharynx, excretory cells (Xiao and Xu, 2009) | 46.2% identity in 262 residues overlap; Score: 586.0; Gap frequency: 3.8% |

Continued

**Table 1. Continued**

| Gene name | Function | O. vulgaris candidate | C. elegans orthologue | Allele | Strain name | Mutation | Reported behavioural phenotype | Gene expression | % domain identity |
|---|---|---|---|---|---|---|---|---|---|
| NO Mechanoreceptor Potential C (NOMPC, TRPN) | TRPN channel involved in mechanosensation (Turner et al., 2016; Yan et al., 2013) | c31314_g5_i1 PV164451 | trp-4 | sy695 | TQ296 | Deletion (3 kb). Eliminates sequence in the 3' region (Li et al., 2006) | Abolished CEP neuron mechanoreceptor currents. Insensitivity to proprioception and posterior harsh touch (Li et al., 2006; Li et al., 2011) | Dopaminergic ciliated sensory neurons (CEP, PDE, ADE) and in DVA, DVC interneurons (Xiao and Xu, 2009) | 29.0% identity in 928 residues overlap; Score: 556.0; Gap frequency: 6.6% |
| Transient Receptor Potential P. (TRPP) | Ion channel involved in mechanosensitive transduction signal (Giamarchi et al., 2006) | c34000_g4_i1 PV164454 | pkd-2; him-5 | pkd-2 (sy606) him-5 (e1490) | PT8 | Null mutation producing a truncated protein from the middle of tm1 onward (Barr and Sternberg, 1999) | Chemo- and mechano-defects in male mating behaviour (Barr and Sternberg, 1999) Unknown involvement in nociception | Male-specific mechanosensory/ chemosensory neurons: CEM, HOB, (Barr and Sternberg, 1999) | 44.7% identity in 425 residues overlap; Score: 914.0; Gap frequency: 1.6% |
| Transient Receptor Potential Vanilloid. (TRPV) | Ion channel involved in heat and low pH perception (Caterina et al., 1997; Immke and Gavva, 2006; Liu and Wood, 2011) | c32354_g12_g2_g7_g14_g6 PV164572 | osm-9 / ocr-2 / ocr-2, osm-9, ocr-1 | ky10 / ak47 / ocr-2 (ak47) osm-9 (ky10) ocr-1 (ak46) | CX10 / CX4544 / FG125 | Substitution (Glu/*) Nonsense mutation Deletion (1960 bp) (Colbert et al., 1997; Tobin et al., 2002) ak47: Deletion (1960 bp); ky10: Substitution (Q/*) ak46: Deletion (2026 bp) (Colbert et al., 1997; Tobin et al., 2002) | Insensitivity to nose touch, hyperosmolarity, low pH and aversive volatile (Colbert et al., 1997; Sambongi et al., 2000; Tobin et al., 2002) | Amphid sensory neurons (ASH, AWA, AWC, ASE, ADF, ASI, ASJ, ASK, IL2 s), PVD, OLQ, PHA, PHB), rectal gland cells, uterine cells (Colbert et al., 1997) | 53.7% identity in 214 residues overlap; Score: 618.0; Gap frequency: 0.5% |
| Piezo | Ion channel involved in mechanosensation and its transduction (Mikhailov et al., 2019; Wang and Ragsdale, 2019) | c36392_g4_i1 MN081842 | pezo-1 | av143 | AG405 | Deletion (4 kb). Eliminates from 27th to 33rd exon+ Insertion of stop codon (Komandur et al., 2023) | Altered crawling behaviour. Unknown involvement in nociception (Komandur et al., 2023) | Pharynx, Reproductive tissues, neurons HOB, PCB, PCC, and various ray neurons (Bai et al., 2020; Millet et al., 2021) | 66.0% identity in 53 residues overlap; Score: 197.0; Gap frequency: 0.0% |
| **Gene name** | **Function** | **O. vulgaris candidate** | **C. elegans orthologue** | **Allele** | **Strain name** | **Mutation** | **Reported behavioural phenotype** | **Gene expression** | **% domain identity** |
| **Voltage-gated ion channels/subunits** | | | | | | | | | |
| Calcium channel, voltage-dependent, alpha 2/delta subunit 3. (CACNA2D3) | Calcium channel associated with high heat response - orthologue of Drosophila straightjacket (Neely et al., 2010) | c30247_g1_i1 PV164539 | unc-36 | e251 | CB251 | Substitution (Gly452*), Nonsense mutation leading to a truncated protein in the first third (Lainé et al., 2011) | Defects in touch avoidance behavior (Frøkjær-Jensen et al., 2006) | Mechanosensory neurons (AVM, ALM, PVM, PVQ, PVC, DUC, and DVA) (Frøkjær-Jensen et al., 2006) | 33.3% identity in 60 residues overlap; Score: 111.0; Gap frequency: 1.7% |
| Potassium Two Pore Domain Channel Subfamily K. (KCNK) | TWIK-relate channel triggered by changes in the surrounding pH (Li and Toyoda, 2015) | c33106_g1_i2 PV164533 | twk-46 | tm110925 | - | Deletion (99 bp). Eliminates from the 3rd to the 7th exon (Zhou et al., 2022) | Moderate/severe locomotor impairments. Unknown involvement in aversive sensory detection (Zhou et al., 2022) | AVE, AVB, backward (VA, DA) and forward (VB, DB) excitatory motoneurons and in few inhibitory motoneurons (VD, DD) (Zhou et al., 2022) | 53.5% identity in 129 residues overlap; Score: 356.0; Gap frequency: 5.4% |

Continued

**Table 1. Continued**

| Gene name | Function | O. vulgaris candidate | Allele | Strain name | Mutation | Reported behavioural phenotype | Gene expression | % domain identity |
|---|---|---|---|---|---|---|---|---|
| Potassium Voltage-Gated Channel Subfamily Q. (KCNQ) | K channels involved in modulating pain threshold (Wood, 2006) | c33477_g8_i1 PV164532 c33539_g2_i3 PV164531 | ok732 | RB883 | Deletion (1694 bp). Eliminates most of the channel pore region (Okahata et al., 2019) | Supranormal cold acclimation. Decreased temperature-dependent activity in ADL neurons (Okahata et al., 2019) | Sensory neurons (ASK, ADL), fan and ray sensory neurons of male adults, and intestinal cells (Okahata et al., 2019) | 32.0% identity in 206 residues overlap; Score: 221.0; Gap frequency: 7.3% |
| Potassium Voltage-Gated Channel Subfamily Q. (KCNQ) | | | tm542 | TM542 | Deletion (1001 bp). Eliminates from 2nd to 4th exon (Bauer et al., 2019) | Enhanced thermosensitivity (Okahata et al., 2019) | | 68.4% identity in 367 residues overlap; Score: 1178.0; Gap frequency: 9.0% |
| Potassium Voltage-Gated Channel Subfamily A. (KCNA) | K channel involved in the modulation of touch and pain (Hao et al., 2013) | c30280_g2_i1 PV164509 | ok1581 | RB1392 | Insertion: CTAAATAT Deletion: 632 bp Wormbase | Reduction of action potential frequency in body wall muscles and thus reduced locomotor activity (Gao and Zhen, 2011; Liu et al., 2018) Unknown involvement in aversive sensory detection | Muscles and neurons (Gao and Zhen, 2011; Liu et al., 2018) | 55.9% identity in 188 residues overlap; Score: 527.0; Gap frequency: 2.1% |
| Gene name | Function | O. vulgaris candidate | Allele | Strain name | Mutation | Reported behavioural phenotype | Gene expression | % domain identity |
| **Proteins related to neuropeptides and lipid metabolism** | | | | | | | | |
| Neuroendocrine convertase 1. (PCSK1) | Involved in the processing of enkephalins into their mature form (Johanning et al., 1998) | c28798_g1_i1 PV164566 | gk8 | VC48 | Deletion (2238 bp). Eliminates from 3rd exon to part of the 5th, resulting in a frameshift after Leu185 and a predicted stop codon after 17 non-homologous residues (Salzberg et al., 2014) | Severely defective dendritic arbours in both PVD and FLP neurons, impaired branching and extension of other sensory neurons, interneurons and motorneurons (Dong et al., 2016; Salzberg et al., 2014) | Broadly expressed in the nervous system (PVD, AQR, VC, AIY, DA, DB, ALM, HSN) (Salzberg et al., 2014) | 50.3% identity in 443 residues overlap; Score: 1125.0; Gap frequency: 2.5% |
| Neuroendocrine convertase 2. (PCSK2) | | c32479_g16_i1 PV164565 | n150 | MT150 | Substitution (Gly594Glu). Missense mutation in Proprotein convertase P-domain (Husson et al., 2006; Kass et al., 2001) | Increased sensitivity to acidic pH, restores touch sensitivity caused by other mutations, heat sensitive. Drastic reduction in neuropeptides (Husson et al., 2006; Kass et al., 2001; Wakabayashi et al., 2015) | ASH, ALM, AVM, and PVM, interneurons AVB, AVD, PVC, RIG, and SDQL; and the HSN egg-laying motorneurons (Kass et al., 2001) | 76.5% identity in 446 residues overlap; Score: 1864.0; Gap frequency: 0.4% |
| Neuroendocrine convertase 2. (PCSK2) | | | n588 | MT1218 | Substitution (Glu117Gln). Missense mutation in the Peptidase S8 pro-domain (Husson et al., 2006; Kass et al., 2001) | Body touch sensitivity is greatly diminished (Kass et al., 2001) | | |
| Neuroendocrine convertase 2. (PCSK2) | | | ok979 | VC671 | Deletion (1578 bp). Eliminates catalytic domain (Husson et al., 2006; Kass et al., 2001) | Impaired odour avoidance Drastic reduction in neuropeptides (Husson et al, 2006; Yamazoe-Umemoto et al., 2015) | | |

Continued

**Table 1. Continued**

| Gene name | Function | O. vulgaris candidate | Allele | Strain name | Mutation | Reported behavioural phenotype | Gene expression | % domain identity |
|---|---|---|---|---|---|---|---|---|
| Carboxypeptidase E. (CPE) | Involved in the biosynthesis of enkephalin (Fricker, 1993) | c30963_g3_i1 PV164485 | n476 | MT1071 | Deletion (123 bp). Out of frame deletion producing a truncated protein lacking most of the catalytic domain (Husson et al., 2007) | Thermal avoidance behaviour significantly hampered. Drastic reduction in neuropeptides (Husson et al., 2007; Nkambeu et al., 2021) | Broad nervous expression. (Husson et al., 2007) | 46.8% identity in 408 residues overlap; Score: 898.0; Gap frequency: 1.7% |
| Angiotensin-converting enzyme. (ACE) | Involved in the processing of opioids precursors (Choi et al., 2021) | c29134_g1_i2 PV164476 | tm12662 tm8421 | - - | Deletion (116 bp)+Insertion (8 bp) Deletion (92 bp)+Insertion (6 bp) Wormbase | Unknown involvement in nociception | Hypodermis, vulva and ray precursor cells (Brooks et al., 2003) | 26.4% identity in 580 residues overlap; Score: 473.0; Gap frequency: 5.0% |
| Neprilysin. (NEP) | Key enzyme involved in the degradation of enkephalins (Skidgel and Erdös, 2004) | c34308_g4_i2 MN081826 | by159 | BR2815 | Deletion (1450 bp). Eliminates promoter, 1st and 2nd exon (Spanier et al., 2005) | Uncoordinated pattern of locomotion (Spanier et al., 2005) Unknown involvement in nociception | Pharynx, head interneuron RIH (Spanier et al., 2005) | 35.3% identity in 326 residues overlap; Score: 506.0; Gap frequency: 1.5% |
| | | c32366_g10_i1 MN081824 | pe356 | JN356 | splicing acceptor mutation in the 1st and 2nd exon (Yamada et al., 2010) | Severe defects in olfactory adaptation to benzaldehyde. Unknown involvement in nociception (Yamada et al., 2010) | Muscle, glia, neurons (GLR, AIM, SMB) (Yamada et al., 2010) | 38.4% identity in 664 residues overlap; Score: 1230.0; Gap frequency: 1.5% |
| Tachykinin receptor. (TACR) | Receptor of tachykinins (Deng et al., 2020; Storm et al., 2013) | c31779_g15_i4 MN081857 | ok2886 | VC2171 | Insertion: GGTGATCTATGT Deletion: 755 bp. (Wormbase) | Reduced aversion-resistant ethanol seeking after pretreatment (Salim et al., 2022) | Broad neuronal expression, hypodermal tissue (Barrett, 1999) | 37.9% identity in 293 residues overlap; Score: 490.0; Gap frequency: 2.4% |
| Tachykinin. (Tk) | Neuropeptide involved in nociceptive sensitisation (Nishimori et al., 2013) | c31437_g10_i1 AB037112.1 | ok1469 | RB1340 | Deletion (700 bp). (Wormbase) | Impaired odour habituation (McDiarmid et al., 2015) | AWC, ASI, PHB and BDU neurons and intestine (Nathoo et al., 2001) | 44.7% identity in 38 residues overlap; Score: 56.0; Gap frequency: 5.3% |
| FMRFamide receptor. (FMRFaR) | GPCR potentially involved in anti-nociceptive responses in invertebrates (Kavaliers et al., 1985; Walters, 2018) | c16104_g1_i1 MN081861 | ok3302 | VC2565 | Deletion (591 bp). Eliminates from ending part of 1st exon and the entire 2nd exon (Wormbase) | Defective in locomotor arousal after tap; impairment of chemotaxis when exposed to appetitive olfactory cues (Chew et al., 2018) Unknown involvement in nociception | Head neurons, including ASH, RID, ASK, AIY, and AVK (Chew et al., 2018) | 28.6% identity in 276 residues overlap; Score: 326.0; Gap frequency: 1.1% |

Continued

**Table 1. Continued**

| Gene name | Function | O. vulgaris candidate | C. elegans orthologue | Allele | Strain name | Mutation | Reported behavioural phenotype | Gene expression | % domain identity |
|---|---|---|---|---|---|---|---|---|---|
| FMRFamide peptide. (FMRFa) | Potentially involved in anti-nociceptive responses in invertebrates (Belardetti et al., 1987; Mackey et al., 1987) | c30173_g7_i1 MN081862 | flp-1 | yn2 | NY7 | Deletion (1.1 kb). Eliminates 567 bp of the promoter region up to the 4th exon (Buntschuh et al., 2018; Nelson et al., 1998) | Nose touch defective (7% responsive), insensitive to hyperosmolarity (Buntschuh et al., 2018) | Head interneurons receiving inputs from the ASH neuron. AVK, AVA, AVE, RIG, RMG, AIY, AIA, and M5 (Schinkmann and Li, 1994) | 33.6% identity in 107 residues overlap; Score: 98.0; Gap frequency: 9.3% |
| | | | | yn4 | NY16 | Deletion (2.1 kb). Eliminates upstream and coding regions (Nelson et al., 1998) | Nose touch defective (30% responsive), insensitive to hyperosmolarity (Nelson et al., 1998) | | |
| Allatostatin C / opioid-like receptor. (OPRL/AstCR) | GPCRs involved in antinociception/ analgesia. (Bachtel et al., 2018) | c22802_g3_i1 PV164564 | npr-17 | tm3210 | - | Deletion (209 bp). Eliminates 6th and 7th exon (Wormbase) | Impairment in aversive response to aversive odorant; abolished pumping when exposed to morphine (Mills, 2014) | AVG, ASI and AUA sensory neurons and the PVPs, PVQs, PQR (Harris et al., 2010; Mills, 2014) | 27.9% identity in 240 residues overlap; Score: 171.0; Gap frequency: 6.7% |
| Buccalin-like. (Ast/Op-like peptide) | Opioids are neuropeptides involved in analgesia (Harrison et al., 1994; Stefano and Salzet, 1999) | c26559_g1_g2 PV206714 | nlp-3 | ok2688 | RB2030 | Deletion (1618 bp). Eliminates exons in the 3' region (Wormbase) | Locomotor impairments (Yemini et al., 2013) | Pharyngeal cells (I1-I4, M1, M3, NSM, I6 and, M2), Ach and 5HT neurons, ADF, ASE, ASH, AWB, ASJ, BAG, HSN, VNC, intestine (Nathoo et al., 2001) | 37.1% identity in 67 residues overlap; Score: 19.0; Gap frequency: 0.0% |
| | | | | tm3023 | FX03023 | Deletion (354 bp). Eliminates exons in the 3' region (Wormbase) | Impaired aversive response to octanol (Harris et al., 2010) | | |
| Gene name | Function | O. vulgaris candidate | C. elegans orthologue | Allele | Strain name | Mutation | Reported behavioural phenotype | Gene expression | % domain identity |
| **Other modulators of nociception** | | | | | | | | | |
| Calcitonin Gene-related Peptide Receptor (CGRPR) | Receptor characterising "peptidergic" nociceptors. Involved in pain transmission (Basbaum et al., 2009; Benemei et al., 2009) | c31238_g13_i1 MN081807 | pdfr-1 | ok3425 | CX14295 | Deletion (605 bp). Eliminates from 5th to 7th exon Wormbase | Strong ascr#3 aversion that requires ASI but is completely insensitive to food thickness, severe disruption in crawling behaviour (Luo and Portman, 2021) | Body wall muscle cells, mechanosensory neurons (PLM, ALM, FLP, OLQD and OLQV), chemosensory neurons (PHA and PHB), ring motor neurons (RMED and RMEV) pharyngeal interneuron pair I1 (Janssen et al., 2008). OLL, FLP, AVM, ALM, AVD, PLM, PVM, PVC, PHA, URX, PQR (Luo and Portman, 2021) | 25.3% identity in 265 residues overlap; Score: 200.0; Gap frequency: 6.8% |
| Stomatin (STOM) | Protein found in mechanosensory neurons (Moshourab et al., 2013; Price et al., 2004) | c33141_g1_i2 PV164567 | mec-2 | e75 | CB75 | Substitution (Ala204Thr). Missense mutation altering the N-terminal half of the stomatin-like region (Gu et al., 1996) | Weak to absent light touch sensitivity due to a disruption of a protein-protein interaction (impaired activity of mec-4 and mec-10). Cl-, Na+, benzaldehyde, 2-butanone, isoamyl alcohol chemotaxis defect (Goodman and Schwarz, 2003; Gu et al., 1996; Zhang et al., 2004) | ALM, PVM (Zhang et al., 2004) | 80.1% identity in 201 residues overlap; Score: 835.0; Gap frequency: 0.0% |
| Anoctamin. (ANO) | Cl- channel activated by Ca, highly expressed in small sensory neurons. Reported to be activated by noxious heat over 44°C (Cho et al., 2012) | c31699_g1_i2 MN081804 | anoh-1 | tm4762 | - | Deletion (202 bp). Results in a frameshift and a premature stop codon after aa 17 of the predicted ANOH-1b ORF and removes the start codon of the alternatively spliced ANOH-1a ORF (Li et al., 2015) | In mec-4 and anoh-1 double mutants the removal of necrotic touch neurons (PLML/R) is significantly delayed. Unknown involvement in aversive sensory detection ORF (Li et al., 2015) | Mechanosensory neurons (ALM, PLM, PVM) ORF (Li et al., 2015) | 28.9% identity in 477 residues overlap; Score: 429.0; Gap frequency: 4.0% |

Continued

**Table 1. Continued**

| Gene name | Function | O. vulgaris candidate | C. elegans orthologue | Allele | Strain name | Mutation | Reported behavioural phenotype | Gene expression | % domain identity |
|---|---|---|---|---|---|---|---|---|---|
| Transmembrane Channel. (TMC) | In C. elegans, Required for salt chemosensation and alkaline aversion (Chatzigeorgiou et al., 2013; Wang et al., 2016) | c31137_g8_i2 PV164552 | tmc-1 | ok1859 | RB1546 | Deletion (2025 bp). Eliminates from 10th to 15th exon Wormbase | Strongly defective in the avoidance of NaCl concentrations above 100 mM, reduced avoidance response to sodium acetate and sodium gluconate; no apparent defect in nose touch avoidance (Chatzigeorgiou et al., 2013; Wang et al., 2016) | ASH, ADF, ASE, ADL, AQR, PQR, URX and PHA, PHC body wall muscles and cholinergic neurons (Chatzigeorgiou et al., 2013; Setty et al., 2022; Wang et al., 2016) | 41.0% identity in 61 residues overlap; Score: 124.0; Gap frequency: 0.0% |
| Sodium leak channel non-selective protein. (NALCN) | May contribute to algogen insensitivity as seen in vertebrates (Eigenbrod et al., 2019; Smith et al., 2020; Zhang and Wei, 2024) | c32050_g13_i1 PV164501 | nca-2 | gk5 | VC9 | Deletion (2970 bp). Eliminates from 8th to part of 16th exon Wormbase | Altered response to halothane and impaired locomotion pattern. Unknown involvement in aversive sensory detection (Yeh et al., 2008) | Broad nervous expression (Yeh et al., 2008) | 52.1% identity in 1070 residues overlap; Score: 2702.0; Gap frequency: 3.8% |
| Cyclic Nucleotide Gated Channel (CNG) | In C. elegans, involved in the transduction cascade for the chemosensory detection and thermosensation (Coburn, 1996; Komatsu et al., 1996) | c31964_g8_i1 PV164549 | tax-2 | ks10 | FK100 | Unknown | Unresponsive to temperature or to water-soluble or volatile chemical attractants (Coburn, 1996; Komatsu et al., 1996) | AQR, AWB, AWC, AFD, ASE, ASG, ASK, ASJ, ASI, PQR, URX (Coburn, 1996; Komatsu et al., 1996) | 35.8% identity in 360 residues overlap; Score: 574.0; Gap frequency: 2.8% |
| | | | tax-2 | p671 | PR671 | Substitution (Cys231Arg). Missense mutation in the first membrane-spanning domain, affecting the pore region (Coburn, 1996) | | | |
| | | | tax-4 | p678 | PR678 | Substitution (Gln/*) Nonsense mutation (Komatsu et al., 1996) | | | 64.6% identity in 189 residues overlap; Score: 584.0; Gap frequency: 2.6% |
| | | | tax-2/tax-4 | p671/p678 | BR5514 | p671: Substitution (Cys231Arg) p678:Substitution (Gln/*) Nonsense mutation (Coburn, 1996; Komatsu et al., 1996) | | | - |

For each gene, the first four columns report the gene proposed function based on literature, the O. vulgaris candidate gene (based on the assembled transcriptome Petrosino, 2015 and Petrosino et al., 2022) and assigned NCBI Accession number. Columns 5-10 refer to the orthologue gene of C. elegans, the strain and alleles selected to be tested for chemoaversion impairments and the pre-existing knowledge on the behavioural phenotype of the mutant, as well as the known localisation of the gene in worm tissues. In some cases, a mutation in more than one allele encompassing the same gene was tested. The last column refers to the degree of identity and similarity found between the two species genes when comparing their functional domains (Expasy-SIM tool, BLOSUM62 comparison matrix).

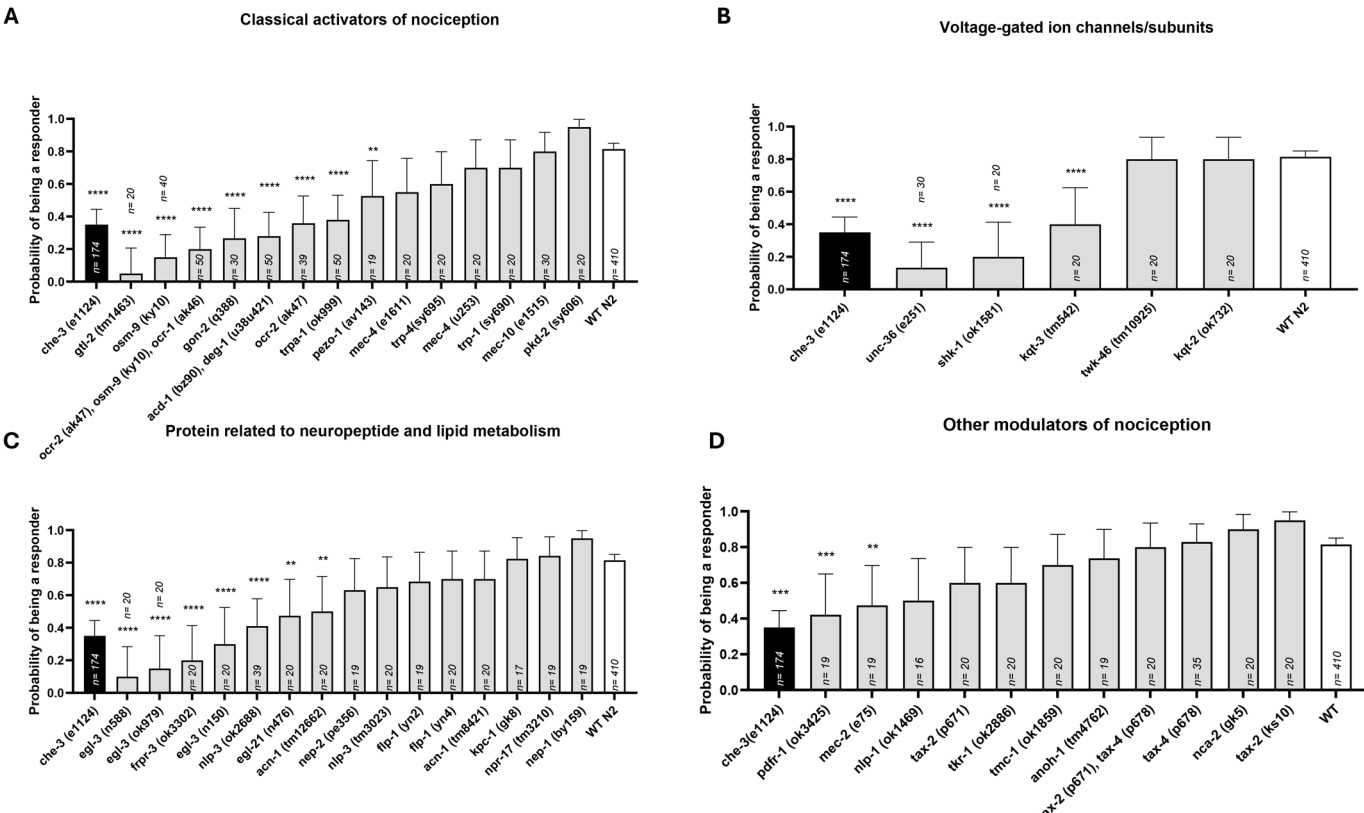

**Fig. 2. Functional investigation of *C. elegans* orthologues of *O. vulgaris* genes implicated in nociception.** The bar graph shows the average predicted probability of being a responder (≥3 reversals within 5 s) and 95% confidence intervals for each strain. The putative genes involved in nociception were grouped into distinct categories (A-D). Category A: "Classical activators of nociception", refers to all the genes that were found recurringly across all the resources we interrogated (e.g. TRPs, ASCs, PIEZOs), Category B: "Ion channel and subunits involved in nociception" is self-explanatory, Category C "Proteins related to neuropeptide and lipid metabolism" was originally meant to include representatives from the opioid and endocannabinoid systems but given we could not find any obvious orthologues, we focused on the enzymes involved in the synthesis, maturation or catabolism or neuropeptides and lipids that might still have a fundamental role for counteracting nociception. Category D: "Other modulators of nociception" refers to a broader range of genes that encode for enzymes and receptors that have been known to be implicated in nociception processing. Significance is expressed according to the corrected *P*-value (q value) following Benjamini-Hochberg false discovery rate method (Q=1%). *n* represents the number of individuals tested. Each strain was tested in at least two independent experiments. The average score and the individual data point is shown in Fig. S2.

*twk-46* (q=0.8707 with an average of 3.2500 reversals, Fig. 3) were altered in their withdrawal response. *unc-36* (q=0.0002), orthologue of the CACNA2D3 subunit, showed a defective response when challenged with the noxious pH3 cue (Fig. 2B and Fig. S2B).

Strains with mutations in genes that encode for 'proteins related to neuropeptide and lipid metabolism', showed defects in the detection of acetic acid (Fig. 2C and Fig. S2C). The *nlp-3 (ok2688)* strain showed a clear impairment in response to the repellent (q=0.0003), whilst mutant strains of FMRFamide-like peptide orthologues (*flp-1*) were responsive. Among the FMRFamide receptors *frpr-3*, showed a loss of response (q= 0.0003, average number of reversals 0.9500, Fig. 3). Considering *C. elegans* orthologues of the enzymes involved in the biosynthesis (*egl-21*), maturation (*kpc-1*, *egl-3*, *acn-1*) and degradation (*nep-1*, *nep-2*) of neuropeptides, the PCSK2 *egl-3* representatives (q=0.0003), the CPE *egl-21* (q=0.0027) and the ACE *acn-1(tm12662)* (q=0.0042) mutant strains showed a reduced response when exposed to the pH3 repellent (Fig. 2C and Fig. S2C).

Belonging to a broader category of 'other nociception modulators', only mutants in the stomatin-like protein *mec-2* and the calcitonin gene-related peptide receptor (CGRP) *pdfr-1*, seemed to be involved in pH 3 avoidance (q= 0.0052 and q=0.0013, respectively). Finally, most of the genes in this category did not show defects in this specific aversive response. This includes CNG

representatives, *tax-2* and *tax-4*, the anoctamin-1 orthologue *anoh-1*, all eliciting a WT-like performance (>3 reversals) when exposed to acetic acid (Fig. 2D, Fig. S2D and Fig. 3).

We additionally tested the worm strains' response to other classical aversive cues such as 4 M fructose, eliciting high osmolarity, 30 mM CuSO$_4$ and 30% 1-octanol as a volatile repellent.

Looking at the soluble cues, a consistent overlap was found for most of the mutant strains that were found to be impaired in the low pH response with a few exceptions (Fig. S3). The mutant *mec-4 (u253)*, which was sensitive to low pH, was found to be significantly impaired to 4 M fructose and 30 mM CuSO$_4$ aversion (q=0.0090 and q=0.0002, respectively). The 'voltage-gated ion channels/ subunits' representative *twk-46 (tm10925)* showed a significant reduction in the responsiveness to 30 mM CuSO$_4$ (q=0.0002) as did the *anoh-1 (tm4762)* strain (q=0.0044).

Few strains were found to be impaired in volatile aversion when exposed to 30% 1-octanol. These include the three TRPV representatives *osm-9 (ky10)*, *ocr-2 (ak47)* and the triple mutant (*P*<0.0001), *gtl-2 (tm1463)* (*P*=0.0006) among the 'classical activators of nociception' category, members of the 'proteins related to neuropeptide and lipid metabolism' such as *flp-1 (yn4)*, *egl-21 (n476)* (*P*<0.0001) and *kpc-1 (gk8)* (*P*=0.0008) and *unc-36 (e251)* as representative of the 'voltage-gated ion channels/subunits' (Fig. S4).

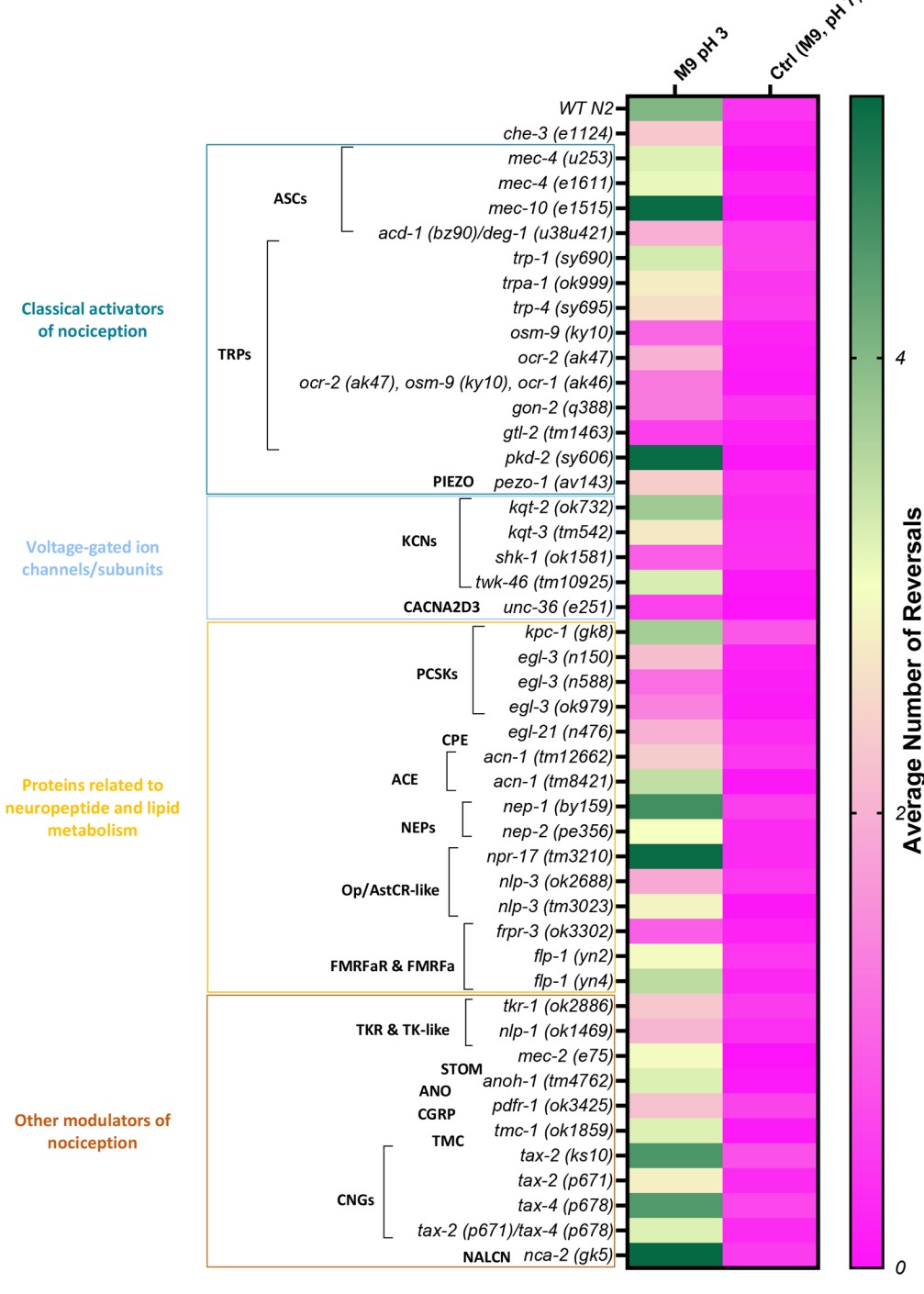

**Fig. 3. *C. elegans* highlights potential molecular determinants involved in *O. vulgaris* nociception.** Data for the drop assay is reported here as the number of actual reversals after exposure to the noxious cue (M9, pH 3). The heat map shows a colour coded indication of the strains' sensitivity to M9, pH 3 from non-sensitive *che-3 (e1124)*-like (magenta) to sensitive WT N2-like (green). Each colour block represents the average number of reversals across all the worms tested per strain. The WT N2 performance was characterised by at least three body reversals within 5 s of contact with the drop and was used to set our criteria for responsiveness.

Altogether, our data shows that most of the strains found to be impaired in the low pH aversion were also defective to other cue modalities (17), only three were exclusively involved in pH (*trpa-1*, *shk-1*, *pdfr-1*) and only four (*mec-4*, *twk-46*, *anoh-1*, *kpc-1*) were found to be involved in responses to non-pH related chemicals. Finally, 12 were found to not be involved at all in any aversive response.

## DISCUSSION
### Core conserved molecular nociceptors in *O. vulgaris* can be identified using functional characterization in *C. elegans*

The *in silico* strategy revealed that a large number of genes showing conservation to known nociceptive genes in other species are present in *O. vulgaris*. In particular, at least two representatives were found to belong to the acid sensing ion channels (ASICs), including the classical molluscan FMRFamide-gated Na$^+$ channel (Cottrell et al., 1983, 1990; Cottrell, 1997), which were proven to be pH sensitive too (Perry et al., 2001), two genes belonging to the PIEZO family, whilst for TRPs, nine members (with multiple isoforms) were identified (Table S2). These include two TRPA1 channels, two TRPCs, two TRPMs, one TRPP, one TRPN and one TRPV channel, showing less variety of members when compared to vertebrates (Peng et al., 2015). This suggests either the vertebrate TRPs underwent specialisation during evolution (e.g. gene duplication and divergence) or that the fewer invertebrate

representatives play multiple physiological functions as in the original ancestral gene (Valencia-Montoya et al., 2024).

When tested in *C. elegans*, one of the strongest behavioural impairments was found in mutant strains for the TRPV channel representatives *osm-9* and *ocr-2*. (Fig. 2A). These results are in line with previous studies showing their role in acidic pH sensation in *C. elegans* (Sambongi et al., 2000) reinforced by the essential role that orthologues play in vertebrate noxious responses (Dhaka et al., 2009). Additionally, in accordance with previous works, these were found to be strongly impaired to all the other cues tested (Figs S3 and S4), showing a conserved polymodal role of this receptor (Radresa et al., 2012) and confirming their overall importance in the primary sensory signalling of *C. elegans* (Colbert et al., 1997; Thies et al., 2016; Tobin et al., 2002).

Interestingly, other TRP channel orthologues were defective in chemoaversion implicating them in the acidic response. These include *trpa-1* and the TRPM representatives *gon-2* and *gtl-2*. The former has been previously found to be implicated in mechanical and thermal responses in *C. elegans* whilst a WT response was elicited when exposed to ASH-dependent noxious chemical cues tested (i.e. copper chloride, 1 M glycerol, denatonium; Kindt et al., 2007) as also shown by our data (Figs S3 and S4). Our results could therefore reflect the complexity of the low pH response circuitry, which does not exclusively rely on ASH neurons but on a series of amphids response (Sambongi et al., 2000; Bergamasco and Bazzicalupo, 2006). To the best of our knowledge, the TRPMs have never been tested for chemical sensitivity but have been mostly studied for their role in reproduction (Sun and Lambie, 1997; Teramoto et al., 2010). Despite the low expression levels of *gon-2* in amphids and the selective expression of *gtl-2* in pharynx and excretory cells, the phenotype observed might indicate some indirect effect of these TRP proteins in the chemosensation executed by exposure to low pH and other cues (Fig. 2A and Fig. S3).

The only ASC strain to show defects when exposed to acidic pH was the double mutant *acd-1/deg-1*, which carries mutation in both neuronal and glial ASC proteins, suggesting a glial role in the processing of low pH in *C. elegans*, similarly to what has been previously reported in vertebrates (Chesler, 2003; Erlichman et al., 2004; Wang et al., 2008; Gourine and Dale, 2022). The octopus arm, the main structure involved in sensory interaction with environmental cues, includes different types of neurons and glial-like cells that might highlight the potential role of this cell type in acidic nociception as well (Young, 1971; Winters-Bostwick et al., 2024). The lack of impairment in the rest of the ASC protein family members tested here is in line with previous studies showing a main role in mechanical detection (Suzuki et al., 2003; Zhang et al., 2004; Árnadóttir et al., 2011; Schafer, 2015; Al-Sheikh and Kang, 2020).

### *O. vulgaris* lacks canonical vertebrate opioid signalling

According to Bateson's (1991) criteria for eligibility to feel pain, animals must be endowed not just with nociceptors connected to a central brain mass, but they should also possess a system able to counteract the inner disruption of the organism homeostasis, namely analgesic pathways. Therefore, we included candidates for opioid and cannabinoid receptors, and their ligand precursors. Blasting of 'classical' vertebrate opioid receptors (i.e. mu-, kappa-, delta-) produced two matches. The first one was the OGFR, which has no structural resemblance to the classical opioid receptors and constitutes a different protein family (Zagon et al., 2002). The second and closest homology was to the Allatostatin C receptor (AstCR), the protostome orthologue of the vertebrate somatostatin receptor (Koch et al., 2022). Opioid receptors are thought to have

evolved from a duplication of the AstCRs (Thiel et al., 2021) and a few studies have reported a role of AstC in nociception and immunity in invertebrates (Bachtel et al., 2018; Li et al., 2021), in addition to growth and feeding regulation (Mizoguchi, 2016; Stay and Tobe, 2007). This raises the intriguing possibility that AstC signalling might underpin anti-nociceptive signalling in phyla that lack an opioid system as supported by recent evidence in *Drosophila melanogaster* (Bachtel et al., 2018; Liu et al., 2023). In line with this, we selected the opioid-like/Allatostatin C-like receptor *npr-17* as the best candidate to be tested in *C. elegans* (Beets et al., 2023). Previous work has shown that the receptor NPR-17 and the neuropeptides NLP-3 and NLP-24 are essential in the morphine-mediated 1-octanol avoidance response and that the impaired response of the *npr-17* null mutant could be rescued by the human κ-opioid receptor (Mills et al., 2016). However, in our experiment, mutant strains for all three genes were not impaired in low pH chemoaversion, confirming a more complex peptidergic modulation of the aversive response. A recent large-scale de-orphanisation of *C. elegans* GPCR identified an additional 12 AstCRs, which might be worth analysing in the appropriate noxious context (Beets et al., 2023).

*C. elegans* orthologues of enzymes involved in the biosynthesis (i.e. the CPE *egl-21*), and processing of neuropeptides (i.e. the PCSK2 *egl-3*), showed disrupted responses to low pH and other cues. Mutations in *egl-3* and *egl-21* were already known to show defects in mechanosensory and thermosensory avoidance responses (Kass et al., 2001; Nkambeu et al., 2019). Such defects can be related to the disrupted production of neuropeptides as highlighted by mass spectrometry experiments (Husson et al., 2006, 2007). Collectively, these data point out the importance of neuropeptide signalling in the processing of nociception (Dickinson and Fleetwood-Walker, 1998) but given the pleiotropic nature of these molecules, does not constitute evidence of the presence of the classical analgesic pathways. Indeed, even in the case of the opioid sensitive analgesic pathway there is a broader functional consequence of this signalling extending to underpinnings of appetite, gut function and systemic homeostasis (Yuan et al., 2012).

Our *in silico* findings are somewhat in line with previous phylogenetic and bioinformatic analysis suggesting opioids and endocannabinoids as vertebrate exclusive proteins (Thiel et al., 2021). This does not necessarily mean that invertebrates do not possess a system to counteract noxious stimuli as they could have more ancient or unique solutions to the same problem. FMRFamide has been considered the equivalent of opioid molecules in invertebrates as the sequence (Phe-Met-Arg-Phe-NH2) shows some similarities to the heptapeptide Met-enkephalin (Tyr-Gly-Gly-Phe-Met-Arg-Phe). Among the wider physiological functions these peptides were found to modulate in molluscs and more generally in invertebrates, there are a few showing an induced suppression of primary nociceptors in *Aplysia* (Belardetti et al., 1987; Mackey et al., 1987).

The mutant of the *C. elegans* FMRFa orthologue we selected, *flp-1*, did not show impairment in acidic response but has been found to be involved in high osmolarity aversion (Nelson et al., 1998; Li et al., 1999), suggesting a more selective chemosensory role. The orthologue of the FMRFamide receptor, *frpr-3*, on the other hand, showed a strong impairment in pH 3 response, and thus represents a strong candidate for further analysis.

### Lipid signalling is an important component of nociception modulation across phyla

In the case of the endocannabinoid pathway, our *in silico* analysis identified different *O. vulgaris* genes encoding proteins implicated

in fatty acid metabolism, such as DAGL, NAPE-PLD or fatty acid-binding proteins (FABPs) (Table S2). However, we did not find molecular homologies to sign post any classical receptors of these signalling cascades within the octopus genomes. Although we found no evidence for the canonical endocannabinoid receptors in octopus, the lipids that act on them have established physiological effects on other proteins such as allatostatin receptors and TRPVs.

Interestingly, one of the proposed 'ancestors' of the endocannabinoid system is the TRPV channel, previously shown to mediate reduced nocifensive response when activated by anandamide and 2-acyl-glycerol in the medicinal leech *Hirudo verbana* (Summers et al., 2017). Specifically in the case of *C. elegans*, bioinformatic analysis coupled with thermal proteomic profile, identified the AstCR *npr-32*, and the TRPV *ocr-2* as responsible for modulating the nocifensive response in a thermal avoidance assay, confirming these molecular determinants as key players in the detection and modulation of nociceptive responses (Abdollahi et al., 2024).

### Limitations and potential confound of the study

Our *in silico* approach was biased towards the selection of pre-existing, conserved molecular determinants, thus excluding potentially phylum-specific proteins akin to the recently identified chemotactile receptors in octopus and squid (van Giesen et al., 2020; Kang et al., 2023). Our study found a large number of conserved candidates in the transcriptome of *O. vulgaris* (Petrosino, 2015; Petrosino et al., 2022). However, the database available is not refined and therefore manual curation led to either the exclusion of incomplete or misannotated transcripts or required manual reconstruction (e.g. TRPV) to be fully investigated. The availability of the recently published chromosome-level genome could potentially improve the quality of this strategy (Destanović et al., 2023). Additionally, BLAST search will limit the findings to sequences with sufficient shared similarities. Using alternative methods such as an HMM-based analysis, could help find candidates with more divergent sequences (Felsenstein and Churchill, 1996).

Despite their phylogenetic distance, with a last common ancestor between Lophotrocozoa and Ecdysozoa around 535 million years ago (Howard et al., 2020), *C. elegans* represents the closest well-characterised (both genetically and phenotypically) model organism to investigate *O. vulgaris* nociceptors. However, the distinct environments (terrestrial versus marine) in which the two species evolved should be considered, as the diverse modalities and variety of ecological stimuli could have driven shared proteins to diverge and acquire different adaptations (van Giesen et al., 2020; Allard et al., 2023).

Additionally, our strategy could not be applied to some genes as they are missing from *C. elegans* genome. In fact, a few *O. vulgaris* gene families representative of nociception signalling could not be modelled in the nematode due to the absence of corresponding orthologues. This is exemplified by the purinergic receptors (P2X) and voltage gated sodium (Na$_v$s) channels (Bargmann, 1998; Harte and Ouzounis, 2002; Hobert, 2005-2018; Burnstock and Verkhratsky, 2009; Table S3).

### Future directions

By performing a chemosensory assay on *C. elegans* mutants for the orthologue of *O. vulgaris* putative nociceptive genes, we have highlighted candidates that are involved in the acidic pH avoidance response. Two main categories of proteins deserve priority for further investigation: one is the classical activators of nociception, such as the TRPV channel subfamily, whose role in nociception is widely conserved across phyla, and the other is the neuropeptide-related proteins, such as FMRFa, AstC, their receptors and the enzymes involved in neuropeptide processing. A functional role of these candidates in nociception could be explored further by complementation of *C. elegans* lost sensory functions by the *O. vulgaris* orthologous gene. Overall, we highlighted an intersecting bioinformatic model hopping approach that facilitates understanding of nociception in *O. vulgaris* and which has key relevance for an evolutionary perspective on the phenomenon of pain.

## MATERIALS AND METHODS
### *In silico* analysis
#### Resources to produce a list of conserved Eumetazoan nociceptive related genes

An in-depth literature review of conserved molecular nociceptors was performed by surveying around 400 peer reviewed journal articles on animal nociception from well-characterised vertebrate and invertebrate Eumetazoa.

This work was complemented with information taken from databases focussed on human pain genetics: 1) Human Pain Genes Database (Meloto et al., 2018) a collection of nociceptive relevant genes resulted from 294 studies reporting 434 associations between genetic variants and pain phenotypes; 2) Pain Research Forum, curated by the International Association for the Study of Pain (IASP), and 3) Pain Genetics MOGILAB (McGill University, Montreal, Canada).

Finally, the queries 'pain perception' and 'opioid activity' were searched within GO (Carbon et al., 2009). The latter was included on the basis that this search term would highlight candidate molecular components that mediate or modulate anti-nociceptive pathways. The results under the 'genes and gene products' label were selected for further analysis.

#### Gene blast against *O. vulgaris* transcriptome and manual curation of sequences

The search for orthologue nociceptive and antinociceptive candidate genes in *O. vulgaris* was performed by iterating the Uniprot amino acid sequence (UniProtKB/Swiss-Prot release 2021_04) against *O. vulgaris* de novo assembly transcriptome (Petrosino, 2015; Petrosino et al., 2022). The transcriptome is based on 64,477 assembled and filtered transcripts with a calculated median length of 795 bp, average length of 1308 bp, minimum length of 201 bp and maximum of 20031 bp (Petrosino, 2015). The assembled *O. vulgaris* transcriptome is available at https://zenodo.org/records/17820138.

The BLAST search was carried out using the TBLASTN algorithm (BLAST+ v2.10.0+). A standard e-value of $10e^{-5}$ was used as a cut-off for the BLAST search except for neuropeptide precursors for which it was set as $10e^{-2}$ and BLOSUM62 as the scoring matrix. The rationale for the latter reduced stringency is due to the known propeptide variation relative to highly conserved and short length mature and active peptide component (Baggerman et al., 2005; Akhtar et al., 2011). When multiple hits were found, we considered the contig sequences with the e-value closest to 0 and a high query coverage (at least 40%) for further analysis. To check for sequence completeness, we performed multiple alignments of our *O. vulgaris* predicted protein and curated sequences from other species using Clustal omega (Madeira et al., 2022). Incomplete sequences from the BLAST search that we failed to reconstruct were discarded. A set of bioinformatic tools was utilised to check the quality of the pre-existing automated annotations by manually curating the sequences. The nearest species with a sequence matching the candidate provided was retrieved from NCBI Blast (Altschul et al., 1990). A prediction of the protein function pathways (GO terms for 'Biological Process', 'Molecular Function' and 'Cellular Component') was obtained via InterProScanSearch (Blum et al., 2021) following translation of the target transcript on Expasy Translate (Gasteiger et al., 2003). The analysis of the conserved domain, the family and shared structure of the protein was also performed, using NCBI conserved domain (Lu et al., 2020) and HHPRed (Hildebrand et al., 2009).

#### Refinement of candidate *O. vulgaris* nociceptive genes

Refinement of the final *O. vulgaris* sequences to be blasted against the *C. elegans* genome was performed by taking into account the available relative gene expression of the candidates in *O. vulgaris* (Petrosino, 2015;

Petrosino et al., 2022). To this end, specific tissues were considered, based on their relevance in neurosensory pathways: TIP, selected due to previous studies showing it is enriched in sensory receptors including those involved in nociceptive responses (Graziadei, 1964; van Giesen et al., 2020); SEM, SUB and OL, constituting the central brain mass fundamental in information processing and thus potentially implicated in the elaboration of nociceptive and anti-nociceptive responses. This process led to the selection of a specific representative for each gene when multiple plausible hits were found.

### Identification of *C. elegans* orthologues for candidate *O. vulgaris* nociceptive genes

*C. elegans* orthologues of the hits derived from the approaches described above, were retrieved from Wormbase resource (Davis et al., 2022) through the available BLASTp tool (version WS283, Bioproject PRJNA13758) using default parameters (threshold e-value: 1e+0).

To compare the degree of molecular conservation between *O. vulgaris* and *C. elegans*, the identity of the protein functional domains was compared with Expasy-SIM tool using BLOSUM62 as the comparison matrix (Huang and Miller, 1991). Where multiple hits were found, the selection was made based on literature evidence for their implication in sensory responses or on their expression in sensory neurons (e.g. *npr-17* as allatostatin C receptor representative within its suggested 13 orthologues; Beets et al., 2023).

### *C. elegans* strains and husbandry

The following nematode strains were utilised in this study: CB1124 *che-3 (e1124)*; CB75 *mec-2 (e75)* [STOM]; TU253 *mec-4 (u253)* [ASIC]; CB1611 *mec-4 (e1611)* [ASIC]; CB1515 *mec-10 (e1515)* [ASIC]; *acd-1/ deg-1 (bz90/u38u421)* [FaNaC]; TQ225 *trp-1 (sy690)* [TRPC]; RB1052 *trpa-1 (ok999)* [TRPA]; TQ296 *trp-4 (sy695)* [TRPN]; CX10 *osm-9 (ky10)* [TRPV]; CX4544 *ocr-2 (ak47)* [TRPV]; FG125 *ocr-2 (ak47), osm-9 (ky10), ocr-1 (ak46)* [TRPV]; EJ1158 *gon-2 (q388)* [TRPM2]; LH202 *gtl-2 (tm1463)* [TRPM3]; PT8 *pkd-2 (sy606)* [TRPP]; AG405 *pezo-1 (av143)* [PIEZO]; RB883 *kqt-2 (ok732)* [KCNQ]; TM542 *kqt-3 (tm542)* [KCNQ]; RB1392 *shk-1 (ok1581)* [KCNA]; *twk-46 (tm10925)* [KCNK]; CB251 *unc-36 (e251)* [CACNA2D3]; VC48 *kpc-1 (gk8)* [PCSK1]; MT150 *egl-3 (n150)* [PCSK2]; MT1218 *egl-3 (n588)* [PCSK2]; VC671 *egl-3 (ok979)* [PCSK2]; MT1071 *egl-21 (n476)* [CPE]; *acn-1 (tm12662)* [ACE]; *acn-1 (tm8421)* [ACE]; BR2815 *nep-1 (by159)* [NEP]; JN356 *nep-2 (pe356)* [NEP]; VC2171 *tkr-1 (ok2886)* [TKR]; RB1340 *nlp-1 (ok1469)* [TK]; VC2565 *frpr-3 (ok3302)* [FMRFaR]; NY7 *flp-1 (yn2)* [FMRFa]; NY16 *flp-1 (yn4)* [FMRFa]; *npr-17 (tm3210)* [AstCR/OPRL]; RB2030 *nlp-3 (ok2688)* [AST/ OP-like]; FX03023 *nlp-3 (tm3023)* [AST/OP-like]; *anoh-1 (tm4762)* [ANO]; CX14295 *pdfr-1 (ok3425)* [CGRPR]; RB1546 *tmc-1 (ok1859)* [TMC]; FK100 *tax-2 (ks10)* [CNG]; PR671 *tax-2 (p671)* [CNG]; PR678 *tax-4 (p678)* [CNG]; BR5514 *tax-2/tax-4 (p671/p678)* [CNG]; VC9 *nca-2 (gk5)* [NALCN]. The Bristol N2 was used as the WT strain.

*C. elegans* hermaphrodite worms were cultured and maintained as described in Brenner (1974). Single colonies of saturated LB broth cultures containing *Escherichia coli* OP50 were used to seed 15 ml nematode growth medium (NGM) agar plates (50 µl per 55 mm diameter plate). Plates containing *C. elegans* were then sealed with parafilm to prevent contamination and kept at 20°C. Three days prior to any assay, gravid adult worms were put on culturing plates to lay eggs for 4 h and then removed to produce an age-synchronised population to be tested at the L4+1 day (young adult) stage. All the experiments have been carried out in compliance with the Ethics and Research Governance Online II (ERGO II) policy (nr 79739) in place at the University of Southampton.

### Drop assay to test acidic aversion in *C. elegans*

To test acidic aversion in *C. elegans* we have used the classical acute aversion test, with modifications (Hilliard et al., 2002, 2004).

On the experimental day, ten L4+1 day old worms were transferred onto a 9 cm unseeded NGM plate (20 ml) and left undisturbed for 20 min, in order to favour the transition from local area search (with high levels of reversal) to dispersal behaviour with low reversals and forward movements (Gray et al., 2005). A small drop of noxious cue was delivered in front of a moving worm through a small glass capillary (1.0 mmOD, ID 0.78 mm, Harvard Apparatus, USA) attached to a 3 ml plastic syringe (Fisherbrand™). The

number of reversals exhibited by the worm within 5 s of exposure to the cue was recorded. A binary score was assigned with a positive response (1) scored for worms reaching the threshold displayed by the WT N2 (at least three complete reversals within 5 s from the exposure to the substance), otherwise a negative response (0) was assigned. Worms for each strain for each condition were tested by exposing them to the drop only once. The resulting response score was calculated and compared to the WT N2 performance. Strains with known or observed strong impaired movements were not included in the study as they could have potentially affected the locomotory readout on which the assay is based.

To trigger low pH response, acetic acid (CAS No. 64-19-7, Fisher Chemicals™) was dissolved in M9 buffer (3 g KH2PO4, 6 g Na2HPO4, 5 g NaCl, 1 ml 1 M MgSO4, to 1 litre H2O; Stiernagle, 2006) to reach a final pH of 3 (M9, pH3).

Additionally, we tested aversion to classical nematode noxious cues such as a 4 M fructose solution (D-Fructose, CAS No. 57-48-7, Sigma-Aldrich) to elicit a high osmolarity response and a solution of 30 mM $CuSO_4$ [Copper(II) sulphate pentahydrate, CAS no. 7758-99-8, ThermoFisher]. Each single cue was compared to the exposure of M9 buffer alone (M9, pH7). All tests were performed at 20°C and the pH of the compounds was checked before administration to the worms. The experimenter was blind to the genotype being tested and exposure to cues was randomised.

Data were analysed using a binomial logistic regression using the WT N2 performance as a reference. Data were corrected using the original false discovery rate methods of Benjamini-Hochberg (Q=1%). Statistics were performed with GraphPad Prism version 10 for Windows (GraphPad Software, Boston, Massachusetts, USA).

### Volatile aversion response in *C. elegans*

In the case of volatile aversion test, using the same setup described for the drop assay, we dipped a thin platinum wire into a 30% 1-octanol solution and passed back and forth it in front of the forward moving animal. The latency (s) to start a first reversal was recorded.

Data were analysed using one-way parametric analysis of variance (ANOVA) with different strains versus WT N2 as between-subjects variable.

Post-hoc comparisons were performed using Dunnett's multiple comparisons A level of probability set at *P*<0.05 was used as statistically significant. Statistics were performed with GraphPad Prism version 10 for Windows (GraphPad Software, Boston, Massachusetts, USA).

### Acknowledgements

We thank Professor Laura Bianchi (University of Miami, Miami, FL, USA) for kindly providing the double mutant *acd-1/deg-1 (bz90/u38)* and Dr Xiaofei Bai (University of Florida, Gainesville, FL, USA) for kindly sharing AG405 *pezo-1 (av143)* strain. The rest of the strains were provided either by the *Caenorhabditis* Genetic Center (CGC), funded by NIH Office of Research Infrastructure Programs (P40 OD010440) or the National Bioresource Project (NBRP), funded by the AMED (Japan Agency for Medical Research and Development).

### Competing interests

The authors declare no competing or financial interests.

### Author contributions

Conceptualization: E.M.P., L.H.-D., G.F., J.D.; Data curation: E.M.P., P.I.; Formal analysis: E.M.P.; Funding acquisition: V.O., L.H.-D., G.F., J.D.; Investigation: E.M.P.; Methodology: E.M.P., V.O., L.H.-D., J.D.; Project administration: V.O., L.H.-D., J.D.; Resources: V.O., L.H.-D., J.D.; Supervision: V.O., L.H.-D., J.D.; Validation: V.O.; Writing – original draft: E.M.P.; Writing – review & editing: E.M.P., V.O., L.H.-D., P.I., G.F., J.D.

### Funding

E.M.P. is funded by the Association for Cephalopod Research 'CephRes' ETS, Napoli, Italy and the HSA-Ceph 1/2019 grant from 'CephRes', and The Gerald Kerkut Charitable Trust. Open Access funding provided by University of Southampton. Deposited in PMC for immediate release.

### Data and resource availability

All relevant data and details of resources can be found within the article and its supplementary information.

**Peer review history**

The peer review history is available online at https://journals.biologists.com/bio/lookup/doi/10.1242/bio.062268.reviewer-comments.pdf

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
