## [Peer Review File · Biology Open]

Identification of molecular nociceptors in *Octopus vulgaris* through functional characterisation in *Caenorhabditis elegans*

Eleonora Maria Pieroni, Vincent O'Connor, Lindy Holden-Dye, Pamela Imperadore, Graziano Fiorito, James Dillon

DOI: 10.1242/bio.062268

Editor: Sandhya Koushika

Review timeline

Original submission:	15 October 2025
Editorial decision:	25 October 2025
First revision received:	16 December 2025
Accepted:	16 December 2025

Original submission

First decision letter

MS ID#: bio.062268

MS Title: Identification of molecular nociceptors in *Octopus vulgaris* through functional characterisation in *Caenorhabditis elegans*

Authors: Eleonora Maria Pieroni, Vincent O'Connor, Lindy Holden-Dye, Pamela Imperadore, Graziano Fiorito, James Dillon

I have now reached a decision on the above manuscript.

The reviewer reports are shown at the bottom of this email or can be accessed, together with a copy of this decision letter, by going to:

As you will see, the reviewers raised a number of substantial criticisms that prevent me from accepting the paper at this stage.

They suggest, however, that a revised version might prove acceptable, if you can address their concerns. Please also note that Biology Open requires that all bar graphs also show individual datapoints, please do replot the graphs (eg Figure 2) including them. If you think that you can deal satisfactorily with the criticisms on revision, I would be pleased to see a revised manuscript. We would then return it to the reviewers.

At this stage, we also ask you to ensure your manuscript complies with our formatting guidelines. Provided you are able to fully address the referees' comments, we are positive about publication of your paper (we accept over 95% of revision submissions) and therefore hope you won't mind any extra work involved in reformatting your manuscript at this point.

Please upload both a 'clean' version of your Word file, along with a highlighted version clearly showing where you have made changes in the revised manuscript. Please avoid using 'Track changes' in Word files as these are lost in PDF conversion.

I should be grateful if you would also provide a point-by-point response detailing how you have dealt with the points raised by the reviewers in the 'Response to Reviewers' box. Please attend to

all of the reviewers' comments. If you do not agree with any of their criticisms or suggestions please explain clearly why this is so.

Reviewer 1

Comments for the author

Referencing databases please cite or add specific releases for databases used. e.g. instead of citing (The Uniprot, 2025) as in line 105 in the manuscript, cite a specific release, using the YYYY_NN format, as recommended by Uniprot <https://www.uniprot.org/help/release-statistics>.

Please also clarify if you've used UniprotKB/Swiss-Prot or UniprotKB/TrEMBL. Examples for citation: UniProtKB/Swiss-Prot release 2017_01 or UniProtKB/TrEMBL release 2020_01.

BLAST method, scoring matrix, parameters BLAST: when performing BLAST searches (Methods, line 107), please specify the specific BLAST software release and version, method (blastx?) and scoring matrix (BLOSUM matrix? which?). Besides the E-value cutoff which is mentioned in the text, which other parameters were used? If all default parameters were used, then specifying the specific BLAST software version is absolutely required.

C. elegans orthologues of O. vulgaris genes

As written, the description in Methods (lines 134-136) of the procedure for finding of orthologues in C. elegans, is confusing or not quite correct. The tool used is BLAST, hence genes were searched for and then, after analysis of BLAST high-scoring pairs (hits), then these hits or presumably the best hit in each case was "retrieved". As mentioned before, please also here specify BLAST algorithm (blastp?), scoring matrix, and other additional parameters that may help readers reproduce the author's results.

Regarding the use of Clustal omega and the ExPasy-SIM tool, again the description and use is confusing since both methods produce alignments. Please clarify the context of use for each: e.g. was Clustal used to obtain global sequence alignments? and the ExPasy-SIM tool to obtain local sequence alignments? and how were these separate alignments produced by these different programs consumed downstream or analyzed?

Drop assay

We would appreciate clarification on several methodological and analytical aspects.

What is the composition or source of the M9 buffer? M9 is used repeatedly throughout the manuscript and no reference, composition or catalog number is provided. Were experiments performed on hermaphrodite worms?

Were experiments performed on hermaphrodite worms?

Since Gray et al. (2025) only describes dispersal behavior, could you please provide the appropriate reference(s) for the Drop acid aversion assay in C. elegans?

What temperature was used to perform the assays? Did you also control the temperature of both the pH 3 and pH 7 solutions?

Were all strains obtained? It would be very helpful to compile all strain information in a table or supplementary table.

It also remains unclear how the "average score" for each drop assay category was calculated. Based on Figures 2 and S2, it appears that all data from independent experiments might have been pooled together. Given that the Methods section states that assays were performed in replicates of approximately 10 animals, an n = 410 for N2 WT suggests that data from independent experiments may have been combined. If that is the case, would this be the most appropriate way to analyze the results, or should the data have been normalized? If the data were indeed pooled, a superplot representation could be considered, as it would allow visualization of independent experiments and make it easier to assess whether the same trend is consistently observed.

Regarding data analysis, we would like to understand why an ANOVA was used for the datasets presented in Figures 2 and S2, given that the main goal appears to be to compare each C. elegans mutant strain lacking a specific gene with the N2 wild type. While it is understandable that all data are shown together for practical reasons, the analysis does not seem to address comparisons among the mutant strains themselves.

Finally, Figures 2 and S2 are presented in different formats. It would be very helpful to include a color code for each C. elegans strain and to use the same type of graph in both figures to facilitate data visualization, at least by category. Are the experimental data shown in Figures 2, 3, and S2 the same?

Bioinformatics pipeline (results)

Results, lines 195-196, "Our bioinformatic pipeline, identified a total number of 474 nociceptive related genes 196 collectively retrieved from literature analysis (141), databases (200) and GO search (133)." As described in Methods and as shown in Figure 1A, the "bioinformatic pipeline" seems to be a number of web searches against the Gene Ontology Resource, downloading of results, followed by manual curation. The definition of a bioinformatics pipeline is "a series of automated computational steps designed to process and analyze biological data" (emphasis on automated). Is this pipeline automated? If so, can the code be made available for readers to reproduce your findings (e.g. with a future Gene Ontology + Uniprot releases?). I know this is a subtle subjective matter but pipeline for bioinformaticians is synonym to code that can be tweaked and run again. Otherwise I'd recommend to authors to call their pipeline a "bioinformatics strategy" instead.

Search of query genes vs *O. vulgaris* transcriptome

Results, lines 215-216: "The search of the 89 genes in *O. vulgaris* transcriptome led to a total number of 228 transcripts." Here the transcriptome is apparently an assembled transcriptome. Otherwise the number of hits (228) seems low to be just raw reads. Please clarify the meaning of transcriptome in this context and maybe also (preferably) describe in the data section of Methods the transcriptome (total number of transcripts, mean, mode, max and min transcript length, and whether splice variants were clustered into a single reference transcript).

MAJOR: the source of the *O. vulgaris* transcriptome is not clearly referenced. For the benefit of readers wishing to reproduce this study, a direct way of accessing the same transcriptome data used should be provided, either in the form of an accession number to a primary databank (EMBL/ENA preferably) or to a secondary permanent data resource (e.g. DataDryad, Zenodo, etc). Currently the bibliographic references provided (Petrosino et al 2015, 2022) are either broken (2015) or provide indirect and convoluted ways of accessing the data. Not sure if this is the author's responsibility or EBI/EMBL's but in any case, it should be made easier for readers to access the data.

As an example, the data for the Petrosino 2022 paper is linked to the ArrayExpress E-MTAB-3957 accession which in turn links to the ENA ERP012773 accession (<https://www.ebi.ac.uk/ena/browser/view/ERP012773>), which does not provide a direct link to the sequences and only allow the view/download of a minimal XML file with links that are also broken and lead to 'Error: 500' technical errors at the EBI/ENA website: please check <https://www.ebi.ac.uk/ena/portal/api/filereport?accession=ERP012773> https://www.ebi.ac.uk/ena/portal/api/filereport?accession=ERP012773&result=read_run&fields=rn_accession,fastq_ftp,fastq_md5,fastq_bytes https://www.ebi.ac.uk/ena/portal/api/filereport?accession=ERP012773&result=read_run&fields=rn_accession,submitted_ftp,submitted_md5,submitted_bytes,submitted_format The same happens when searching the ENA for "ERP012773" as a text string. The results point to Illumina sequencing studies that are again not accessible <https://www.ebi.ac.uk/ena/browser/text-search?query=ERP012773> <https://www.ebi.ac.uk/ena/browser/view/ERX1139396> <https://www.ebi.ac.uk/ena/browser/view/ERR1059208> Accessing all XML files linked also lead to links that lead to 500: error. This should be fixed. Otherwise no reader would be able to reproduce the results of this manuscript Accessing all XML files linked also lead to links that lead to 500: error. This should be fixed. Otherwise no reader would be able to reproduce the results of this manuscript!!

Genes implicated in canonical analgesic pathways "Out of the 89 selected genes, 38 were not found in *O. vulgaris* transcriptome" (line 235). From reading the Ms I understand the lack of identification of similar sequences in the transcriptome may be due to i) lack of representation in the transcriptome of the sourced *O. vulgaris* tissues and specimens; or 2) lack of sufficient sequence similarity which would obscure BLAST detection. In this case maybe an alternative bioinformatics strategy can be devised using profiles as queries (sequence or HMM profiles derived from multiple sequence alignments)

C. elegans mutants of *O. vulgaris* orthologue genes

lines 258-259, "we 258 selected low pH (M9, pH 3) as the exemplar cue..." What does M9 mean in this context? Please clarify.

Discussion, limitations ...

lines 447-449, "The availability of the recently published chromosome-level genome could potentially improve the quality of this pipeline (Destanović et al., 2023)." Here recently means October 2023. While not asking the authors to redo their complete analysis, maybe only do a quick reverse BLAST search to discuss the lack of nociceptive orthologues with some more support? (e.g. use the new DNA genome sequence as query and to a BLASTx against your initial list of genes retrieved from the Gene Ontology Resource?) lines 455-456 "Furthermore, both the phylogenetic distance and the distinct environments in which the two species evolved should be considered..." Here if the authors may accept a suggestion, both *O. vulgaris* (Spiralia) and *C. elegans* (Ecdysozoa) are Protostomes, and after the Ecdysozoa / Spiralia split, *C. elegans* is the closest model organism with very rich omics resources, including mutant strains with well-characterized phenotypes. This may be mentioned to better put in context the choice of *C. elegans* for further experimental work. Finally, another limitation of this work is that a single technique and a single parameter were used to challenge computational screening. When assessing nociceptive responses in *C. elegans*, other experimental approaches are available, such as assays using glycerol or CuSO₄, among others (<https://doi.org/10.1016/j.cub.2022.11.012> <https://doi.org/10.7554/eLife.11642>, <https://doi.org/10.7554/eLife.11642> Drawing conclusions based solely on one experimental technique represents a limitation of the study that should be discussed.

References

The cited reference for the *O. vulgaris* transcriptome is a key one. However, the citation is not for a paper, but for a communication? to the Italian Society of Bioinformatics (BITS). In any case, the reference cannot be accessed. 1) please fix the URL in reference for Petrosino et al 2015. The url is broken in the manuscript, with white space characters after dots; 2) after so the apparently correct URL (https://bioinformatics.hsanmartino.it/bits_library/library/01652.pdf) leads to a server error: Server Error 502 - Web server received an invalid response while acting as a gateway or proxy server. There is a problem with the page you are looking for, and it cannot be displayed. When the Web server (while acting as a gateway or proxy) contacted the upstream content server, it received an invalid response from the content server.

Reviewer 2

Comments for the author

The study by Pieroni et al., ventures into finding molecular candidates for nociceptors in *Octopus vulgaris* by using both bioinformatics and functional characterization. The authors scout several resources of eumetazoan nociceptors, map them against the octopus transcriptome and identify the genes that have orthologs in the genetically tractable organism *C.elegans*. Using several lines of *C.elegans* in which the candidate genes are mutated, the authors conduct a behavioral study examining aversive turning behavior in the nematodes evoked by an acidic stimulus. This way, they narrow their search down to some genes they can prioritize for functional follow up in the *Octopus* itself.

The authors did a great job at collecting the information on potential conserved nociceptor genes and well laid out the bioinformatics pipeline. Some of their finding, e.g. the absence of major opioid pathways from the octopus's transcriptome are particularly interesting and pose evolutionarily relevant questions. It is refreshing to read research on such interesting non-model species, and I am excited to see more of this research, particularly given its advanced cognitive capabilities and the implications of pain for these animals.

Some minor comments: It could be useful to see some information about the relevance of pH mediated avoidance behavior in octopus, as that is the main functional follow up in the manuscript. And I found a typo on p2. 24, cephalopods'.

Another point, it could be interesting to write the divergence time between nematodes and cephalopods, at some point in the manuscript, to put their evolutionary divergence into context (and potentially a phylogenetic tree?); and some more examples of nociception in more closely related species to the octopus (e.g. are there clear orthologs of these in other mollusks, if there is any information on that).

I have one major comment regarding the analysis of the *C.elegans* mutant behavior assay. I will follow up in more detail below, but as a summary, I think the statistical analysis of the scored behavioral data could be refined, and the data visualization might be improved to convey the findings more clearly.

Experimental quality

a) Does each figure have the proper controls?

Yes, controls are used properly.

b) Are experiments performed using appropriate methods that will answer the question (or test the hypothesis or support the observations) posed by the authors? Is the right tool used for the job?

The author's first question was to identify potential gene candidates in *O.vulgaris* that could be interesting for further research on octopus' nociception via an integrated computational pipeline. The authors succeed in laying out how they narrowed down their search to reach the final gene candidate. The authors are thorough in explaining their criteria and mapping the genes onto the octopus transcriptome.

The functional follow-up experiments performed are appropriate to support the author's claim that some of these genes that have orthologs in *C.elegans* are involved in acid-evoked nociception in the nematode. The authors use a drop assay to test turning behavior of individual nematodes to assess how mutations of certain proteins change their aversion to the acidic stimuli. Since pH is only one cause/activator of nociceptors, it would be intriguing and elevating the study to use a different assay in addition to test other forms of nociception (chemical, thermal, mechanical) for their further investigation in the octopus. However, the study succeeds in presenting the tested *O.vulgaris* ortholog and their involvement in nociception even without expanding into other regimes of nociception.

c) Were the data analyzed using appropriate statistical tests?

My main comment on the revision of this paper is the data in Figure 2. While I am convinced of the difference in aversion behavior of different mutant strains and their involvement in nociception, it's not fully clear to me why the behavior was scored as 0/1, since it could also be analyzed and presented as the number of reversals in the 5 s post-exposure (just as in figure 3). If the authors want to continue with the scoring system, I suggest the authors reconsider the data presentation and statistical analysis.

In Figure 2, the statistical test used by the authors is a parametric ANOVA. Since the variable is binary (0,1) rather than continuous, a parametric ANOVA may not be optimal. The analysis here rather compares a probability that a nematode strain is highly responsive/avoidant to the stimulus (score 1, many reversals) vs. not highly responsive (score 0). So I think a binomial GLM (logistic regression) with a Benjamini-Hochberg or Holm-Bonferroni correction would be more appropriate, since the question is whether a mutation changes the probability of exhibiting frequent reversals, rather than comparing the averages. The wild type probability should be your reference in the model and you can check how likely it is that mutations change the probability for the worms to reverse frequently.

As for the representation in Figure 2, the continuous scale on the y axis is misleading, since there is no other value than 0 or 1 in this analysis. So rather than reporting the means and SD which go beyond 1, I would suggest reporting the probability of being a frequent turner (score 1) based on the binomial model and indicate +/- 95% confidence interval around it. This should make the values not exceed 0 and 1.

As for the representation in Suppl. Fig2, the representation of the binary with the violin plots does also not seem like the best way to present the data to me, since it is discrete and binary and not a real distribution that needs to be smoothed out by a violin plot. It suggests there is data between that does not exist. An alternative visualization—such as a swarmplot, stripplot, or stacked barplot—might convey the discrete nature of the data more effectively.

Reviewer's Responses to Questions

Experimental quality

Does each figure have the proper controls?

If 'No', please indicate reasons in Comments for Author box below.

Reviewer #1:

- Yes

Reviewer #2:

- Yes
-

Were the data analyzed using appropriate statistical tests?

If 'No', please indicate reasons in Comments for Author box below.

Reviewer #1:

- No

Reviewer #2:

- No
-

Reproducibility

Were experiments performed using adequate number of biological replicates?

If 'No', please indicate reasons in Comments for Author box below.

Reviewer #1:

- Yes

Reviewer #2:

- Yes
-

Does the methods section provide sufficient detail to permit reproducibility?

If 'No', please indicate reasons in Comments for Author box below.

Reviewer #1:

- No

Reviewer #2:

- Yes
-

Completeness

Are the manuscript's conclusions supported by the data?

If 'No', please indicate reasons in Comments for Author box below.

Reviewer #1:

- Yes

Reviewer #2:

- Yes
-

Scholarship

Do the authors cite and discuss the merits of data that would argue for and against their conclusion?

If 'No', please indicate reasons in Comments for Author box below.

Reviewer #1:

- Yes

Reviewer #2:

- Yes
-

Does the manuscript title & abstract accurately reflect the contents of the manuscript, without hyperbole?

If 'No', please indicate reasons in Comments for Author box below.

Reviewer #1:

- Yes

Reviewer #2:

- Yes

First revision

Author response to reviewers' comments

We Thank the Editor for their effort in mediating the interactions with the Reviewers. We would like to start by addressing the Editor's point regarding the individual data points representation. In the original submission, our data (Figure 2) were represented as a binary score and therefore we believed showing the individual data points would have not been very helpful in representing the data distribution. We had therefore left the original representation for the main text and provided a supplementary Figure (Figure S2) with the requested individual data points.

Following Reviewers' suggestions on statistical analysis and data representation (see below), we kept this formatting in Figure 2, now representing the average probability of being a responder (strain) while Figure S2 shows the average score with individual data points. The same criterion has been applied to Supplementary Figure S3.

Figure S1 and Figure S4 show the individual data points instead, as they are not represented as binary scores.

We thank both Reviewers for their words of appreciation and strong engagement with the submission. Their feedback has helped improve this manuscript and we indicate below how we have addressed the specific comments of Reviewer 1 and 2 respectively.

In this we show the nature of the specific query in unbolded font and illustrate our response in bolded font. In addition, we cross reference using the line numbers to where these changes might appear in the modified submission.

Comments from the Reviewers

Reviewer 1:

- Referencing databases please cite or add specific releases for databases used. e.g. instead of citing (The Uniprot, 2025) as in line 105 in the manuscript, cite a specific release, using the YYYY_NN format, as recommended by Uniprot <https://www.uniprot.org/help/release-statistics>.

Please also clarify if you've used UniprotKB/Swiss-Prot or UniprotKB/TrEMBL. Examples for citation: UniProtKB/Swiss-Prot release 2017_01 or UniProtKB/TrEMBL release 2020_01.

We have now added the level of details and clarifications requested (UniProtKB/Swiss-Prot release 2021_04) **on line 108-109**

- BLAST method, scoring matrix, parameters BLAST: when performing BLAST searches (Methods, line 107), please specify the specific BLAST software release and version, method (blastx?) and scoring matrix (BLOSUM matrix? which?). Besides the E-value cutoff which is mentioned in the text, which other parameters were used? If all default parameters were used, then specifying the specific BLAST software version is absolutely required.

This level of detail (BLAST version, method, matrix etc.) has been now added **on lines 114-116**

- C. elegans orthologues of O. vulgaris genes
As written, the description in Methods (lines 134-136) of the procedure for finding of orthologues in C. elegans, is confusing or not quite correct. The tool used is BLAST, hence genes were searched for and then, after analysis of BLAST high-scoring pairs (hits), then

these hits or presumably the best hit in each case was "retrieved". As mentioned before, please also here specify BLAST algorithm (blastp?), scoring matrix, and other additional parameters that may help readers reproduce the author's results.

We have utilised the Blast-p tool that is incorporated within the Wormbase platform (https://wormbase.org/tools/blast_blat). The versions were already included in the original manuscript, but we have now added the specification that it was a Blast-p search with default parameters (lines 146-147).

- Regarding the use of Clustal omega and the Expassy-SIM tool, again the description and use is confusing since both methods produce alignments. Please clarify the context of use for each: e.g. was Clustal used to obtain global sequence alignments? and the Expassy-SIM tool to obtain local sequence alignments? and how were these separate alignments produced by these different programs consumed downstream or analyzed?

This represents an oversight when writing the manuscript. Thank you for noticing this mistake. Yes, clustal omega was used for global alignments when comparing multiple sequences to check for the *O. vulgaris* hits for completeness. Aligning with curated sequences from other species in some cases helped us checking if there were missing amino acids in the predicted proteins. We have now moved this in the appropriate section (lines 120-122)
Expassy SIM tool was utilised as a local alignment tool between *O. vulgaris* and *C. elegans* orthologue sequences to get the percentage of identity that we have then reported in Table 1 (lines 151-152).

- Drop assay
 We would appreciate clarification on several methodological and analytical aspects. What is the composition or source of the M9 buffer? M9 is used repeatedly throughout the manuscript and no reference, composition or catalogue number is provided.

We have now updated the section with the composition of M9 buffer and appropriate reference (line 198-199)

- Were experiments performed on hermaphrodite worms?

We have now specified we utilised hermaphrodite worms (line 170)

- Since Gray et al. (2025) only describes dispersal behavior, could you please provide the appropriate reference(s) for the Drop acid aversion assay in *C. elegans*?

Gray et al. 2005, was cited to specify our criteria for checking for any confounders caused by *C. elegans* natural tendency of spontaneous reversals on unseeded plates within a definite time frame post transfer and was not intended to be used as a reference for the actual test. We have now added the references to the classical drop assay in lines 181-182

- What temperature was used to perform the assays? Did you also control the temperature of both the pH 3 and pH 7 solutions?

We have now specified that our assay has been carried out at lab temperature (20 °C) and that the solutions were checked for their pH and temperature prior the test (lines 204-205).

- Were all strains obtained? It would be very helpful to compile all strain information in a table or supplementary table.

A statement of where all the strains come from was presented in the original manuscript in the acknowledgments section (lines 537-542). Strain information on allelic mutation, cellular expression and reported phenotype were available in Table 1 of the original version of the manuscript.

- It also remains unclear how the "average score" for each drop assay category was calculated. Based on Figures 2 and S2, it appears that all data from independent experiments might have been pooled together. Given that the Methods section states that assays were performed in replicates of approximately 10 animals, an $n = 410$ for N2 WT suggests that data from independent experiments may have been combined. If that is the case, would this be the most appropriate way to analyze the results, or should the data have been normalized? If the data were indeed pooled, a superplot representation could be considered, as it would allow visualization of independent experiments and make it easier to assess whether the same trend is consistently observed.
- Regarding data analysis, we would like to understand why an ANOVA was used for the datasets presented in Figures 2 and S2, given that the main goal appears to be to compare each *C. elegans* mutant strain lacking a specific gene with the N2 wild type. While it is understandable that all data are shown together for practical reasons, the analysis does not

seem to address comparisons among the mutant strains themselves.

Finally, Figures 2 and S2 are presented in different formats. It would be very helpful to include a color code for each *C. elegans* strain and to use the same type of graph in both figures to facilitate data visualization, at least by category. Are the experimental data shown in Figures 2, 3, and S2 the same?

Thank you for the very valuable feedback. Following the suggestions from Reviewer 2, we have now redone the statistical analysis using a binomial GLM (logistic regression) with a Benjamini-Hochberg correction (lines 207-209).

As for the representation in Figure 2, we have again followed the advice from Reviewer 2 suggesting we report the probability of being a responder based on the binomial model and the 95% CI around the mean to contain the y-axis values within the 0-1 range.

As for Data in Figure 2 and S2, the data is the same but was represented with individual data points following a request from the journal. We have modified the figure S2 according to the suggestion from Reviewer 2 using a scatter dot plot rather than a violin plot which would have given a false impression of continuous data distribution when it is in fact binary.

- Bioinformatics pipeline (results)
Results, lines 195-196, "Our bioinformatic pipeline, identified a total number of 474 nociceptive related genes 196 collectively retrieved from literature analysis (141), databases (200) and GO search (133)." As described in Methods and as shown in Figure 1A, the "bioinformatic pipeline" seems to be a number of web searches against the Gene Ontology Resource, downloading of results, followed by manual curation. The definition of a bioinformatics pipeline is "a series of automated computational steps designed to process and analyze biological data" (emphasis on automated). Is this pipeline automated? If so, can the code be made available for readers to reproduce your findings (e.g. with a future Gene Ontology + Uniprot releases?). I know this is a subtle subjective matter but pipeline for bioinformaticians is synonym to code that can be tweaked and run again. Otherwise I'd recommend to authors to call their pipeline a "bioinformatics strategy" instead. Search of query genes vs *O. vulgaris* transcriptome

We thank the reviewer for the clarification. We have indeed carried out a bioinformatic strategy rather than a bioinformatic pipeline *sensu stricto*. We have therefore made this clearer in the text (lines 13, 24, 223, 379, 506).

- Results, lines 215-216: "The search of the 89 genes in *O. vulgaris* transcriptome led to a total number of 228 transcripts." Here the transcriptome is apparently an assembled transcriptome. Otherwise the number of hits (228) seems low to be just raw reads. Please clarify the meaning of transcriptome in this context and maybe also (preferably) describe in the data section of Methods the transcriptome (total number of transcripts, mean, mode, max and min transcript length, and whether splice variants were clustered into a single reference transcript).

Yes, the transcriptome we are referring to is indeed the result of a *de novo* assembly and therefore the resulted hits are not corresponding to raw reads.

- Please note that the previous Petrosino et al., 2015 reference included in the original submission, was a short note linked to an extensive abstract published after BITS conference where we presented the results of our transcriptome and analysis for the first time. We have now replaced the reference with the actual PhD submission which is publicly available (<http://www.fedoa.unina.it/10212/>). Here, all the statistics requested by the Reviewer (total number of transcripts, mean, mode, max and min transcript length) can be found. We have also added these details to the text (lines 109-113).

The final curated transcripts (including any splice variants) were already available in Table S2 of the original manuscript.

We have also revised and added more detail on Table S1B on the unique representatives from different subfamilies, the total complete and incomplete sequences that led to the final number of 171 transcript registered on NCBI (as per Table S2).

- MAJOR: the source of the *O. vulgaris* transcriptome is not clearly referenced. For the benefit of readers wishing to reproduce this study, a direct way of accessing the same transcriptome data used should be provided, either in the form of an accession number to a primary databank (EMBL/ENA preferably) or to a secondary permanent data resource (e.g. DataDryad, Zenodo, etc). Currently the bibliographic references provided (Petrosino et al

2015, 2022) are either broken (2015) or provide indirect and convoluted ways of accessing the data. Not sure if this is the author's responsibility or EBI/EMBL's but in any case, it should be made easier for readers to access the data.

As an example, the data for the Petrosino 2022 paper is linked to the ArrayExpress E-MTAB-3957 accession which in turn links to the ENA ERP012773 accession (<https://www.ebi.ac.uk/ena/browser/view/ERP012773>), which does not provide a direct link to the sequences and only allow the view/download of a minimal XML file with links that are also broken and lead to 'Error: 500' technical errors at the EBI/ENA website: please check <https://www.ebi.ac.uk/ena/portal/api/filereport?accession=ERP012773> https://www.ebi.ac.uk/ena/portal/api/filereport?accession=ERP012773&result=read_run&fields=run_accession,fastq ftp,fastq_md5,fastq_bytes https://www.ebi.ac.uk/ena/portal/api/filereport?accession=ERP012773&result=read_run&fields=run_accession,submitted ftp,submitted_md5,submitted_bytes,submitted_format The same happens when searching the ENA for "ERP012773" as a text string. The results point to Illumina sequencing studies that are again not accessible <https://www.ebi.ac.uk/ena/browser/text-search?query=ERP012773> <https://www.ebi.ac.uk/ena/browser/view/ERX1139396> <https://www.ebi.ac.uk/ena/browser/view/ERR1059208> Accessing all XML files linked also lead to links that lead to 500: error. This should be fixed. Otherwise no reader would be able to reproduce the results of this manuscript Accessing all XML files linked also lead to links that lead to 500: error.

This should be fixed. Otherwise no reader would be able to reproduce the results of this manuscript!!

We thank the Reviewer for raising this important point. We would like to clarify that the *O. vulgaris* transcriptome used in our analyses is the one previously assembled, annotated and published by Petrosino (2015), **a PhD thesis submission**, and Petrosino et al. (2022; see also Imperadore et al., 2023).

The previous Petrosino et al., 2015 reference included in the original manuscript, was a short note linked to an extensive abstract published after BITS conference where we presented the results of our transcriptome and analysis for the first time. We have now replaced the reference with citation to the actual PhD submission (<http://www.fedoa.unina.it/10212/>), on the basis of which the available assembly has been generated.

To the best of our knowledge and verifications, the datasets and accession numbers provided in Petrosino et al., 2022 remain accessible through the repositories:

- The transcriptome data of *O. vulgaris* are available as downloadable fasta file in additional file 2 in Petrosino et al., 2022
- Data of reads including also arm samples that was used to make the assembly are available at: <https://www.ebi.ac.uk/ena/browser/view/ERP012773>

It is possible the technical difficulties experienced by the Reviewer were due to the recent US administration shutdown causing temporary malfunction of the public repository. At the time of writing this response, the links appear to be fully functional.

Finally, to facilitate the accession to the available transcriptome we have proceeded in uploading the related fasta file on Zenodo with the following DOI: <https://doi.org/10.5281/zenodo.17820138>

This has been added directly to the materials and methods **(line 112-113)**.

- Genes implicated in canonical analgesic pathways "Out of the 89 selected genes, 38 were not found in *O. vulgaris* transcriptome" (line 235). From reading the Ms I understand the lack of identification of similar sequences in the transcriptome may be due to i) lack of representation in the transcriptome of the sourced *O. vulgaris* tissues and specimens; or 2) lack of sufficient sequence similarity which would obscure BLAST detection. In this case maybe an alternative bioinformatics strategy can be devised using profiles as queries (sequence or HMM profiles derived from multiple sequence alignments)

We have addressed some of the limitations in the appropriate section of the discussion, we have now included also the limitation of only using BLAST search as the chosen strategy **(507-509)**.

- *C. elegans* mutants of *O. vulgaris* orthologue genes lines 258-259, "we 258 selected low pH (M9, pH 3) as the exemplar cue..." What does M9 mean in this context? Please clarify.

As above mentioned, M9 composition and clarification has been put in the relevant method section **(line 198-199)**.

- Discussion, limitations ...
lines 447-449, "The availability of the recently published chromosome-level genome could potentially improve the quality of this pipeline (Destanović et al., 2023)." Here recently means October 2023. While not asking the authors to redo their complete analysis, maybe only do a quick reverse BLAST search to discuss the lack of nociceptive orthologues with some more support? (e.g. use the new DNA genome sequence as query and to a BLASTx against your initial list of genes retrieved from the Gene Ontology Resource?)

We thank the Reviewer for the comment. This made us realise that some of the representatives which have expanded members in vertebrates were not pooled together, thus leading to an overestimation of the missing members. We have improved Table S1B to show exactly what we have done. By updating this, the total number of genes (as unique representatives of a protein family) was 74. Out of this, 21 were missing, while 53 were found with at least one representative.

Following this, we have performed the reverse BLAST search of the 21 missing representatives in the new genome of *O. vulgaris*. The search led similar results and identified one representative PRDM12, not identified in the original submission.

lines 455-456 "Furthermore, both the phylogenetic distance and the distinct environments in which the two species evolved should be considered..." Here if the authors may accept a suggestion, both *O. vulgaris* (Spiralia) and *C. elegans* (Ecdysozoa) are Protostomes, and after the Ecdysozoa / Spiralia split, *C. elegans* is the closest model organism with very rich omics resources, including mutant strains with well-characterized phenotypes. This may be mentioned to better put in context the choice of *C. elegans* for further experimental work.

We thank the reviewer for the extremely valid point. We have incorporated this in the text (lines 510-513).

Finally, another limitation of this work is that a single technique and a single parameter were used to challenge computational screening. When assessing nociceptive responses in *C. elegans*, other experimental approaches are available, such as assays using glycerol or CuSO₄, among others (<https://doi.org/10.1016/j.cub.2022.11.012> <https://doi.org/10.7554/eLife.11642>, <https://doi.org/10.7554/eLife.11642>) Drawing conclusions based solely on one experimental technique represents a limitation of the study that should be discussed.

We did test a larger set of compounds including high osmolarity representative cues (4M fructose), volatile repellents (30% 1-octanol) and 30 mM CuSO₄ as additional cues and these have now been added as supplementary (Figs. S3-S4) to the manuscript and we modified the main text (introduction- line 83; methods - lines 200-203, 212-220; results - lines 355-374, discussion - lines 396-400, 404-406, 414, 454-455, 460-462) accordingly. However, the inclusion of only low pH in the original manuscript was based on two main reasons:

- 1) low pH is one of the few nociceptive cues known to trigger a response in cephalopods and therefore relevant in their environmental context
- 2) Addressing a response to each of the cue we believed would have detracted from the main message. Namely, *C. elegans* is a valuable platform filter for relevant genes meriting further functional characterisation. This would include some of the follow up analysis of their role in more than a chemosensory response or modality.

References

The cited reference for the *O. vulgaris* transcriptome is a key one. However, the citation is not for a paper, but for a communication? to the Italian Society of Bioinformatics (BITS). In any case, the reference cannot be accessed. 1) please fix the URL in reference for Petrosino et al 2015. The url is broken in the manuscript, with white space characters after dots; 2) after so the apparently correct URL

(https://bioinformatics.hsanmartino.it/bits_library/library/01652.pdf) leads to a server error: Server Error 502 - Web server received an invalid response while acting as a gateway or proxy server. There is a problem with the page you are looking for, and it cannot be displayed. When the Web server (while acting as a gateway or proxy) contacted the upstream content server, it received an invalid response from the content server.

We have addressed this under the Reviewer's MAJOR comment above.

Reviewer 2:

- Some minor comments: It could be useful to see some information about the relevance of pH mediated avoidance behavior in octopus, as that is the main functional follow up in the manuscript. And I found a typo on p2. 24, cephalopods'.

In the original submission we were referencing to works that used acetic acid, but we have now explicitly mentioned and described the reactions of the animals to its exposure (lines 45-51).

We have also corrected the typo. Thank you for noticing.

- Another point, it could be interesting to write the divergence time between nematodes and cephalopods, at some point in the manuscript, to put their evolutionary divergence into context (and potentially a phylogenetic tree?); and some more examples of nociception in more closely related species to the octopus (e.g. are there clear orthologs of these in other mollusks, if there is any information on that).

We have added some information on the evolutionary distance between Lophotrochozoa (to which molluscs belong) and Ecdysozoa (to which nematodes belong) to both justify the choice of *C. elegans* despite the distance but also to point out that the evolutionary divergence, especially related to their different environmental contexts, should still be considered (lines 510-516).

Regarding the orthologues of these genes in other molluscs, there is mostly bioinformatic evidence showing that these classes are indeed conserved and found in other molluscs too.

However, details on their role in nociception in molluscs has not been investigated in depth.

The original submission mentioned some studies that showed FMRFamide involved in noxious suppression in gastropods (lines 469-474). We also mentioned the finding of the FaNaCs which seem to be molluscan-specific amiloride sensitive ligand gated ion channels. We have specified these have been found to be pH sensitive too (line 380-384) but to the best of our knowledge it was not investigated in a nociceptive context.

- I have one major comment regarding the analysis of the *C. elegans* mutant behavior assay. I will follow up in more detail below, but as a summary, I think the statistical analysis of the scored behavioral data could be refined, and the data visualization might be improved to convey the findings more clearly.

We thank the reviewer for the suggestions, and have addressed them in the sections below.

- Experimental quality

a) Does each figure have the proper controls?

Yes, controls are used properly.

Each experiment was paired such that a wildtype cohort were tested and interleaved with the mutant for each run. With respect to test compound M9, pH 3 all experiments were run against M9, pH 7, the non nociceptive control cue.

b) Are experiments performed using appropriate methods that will answer the question (or test the hypothesis or support the observations) posed by the authors? Is the right tool used for the job?

The author's first question was to identify potential gene candidates in *O. vulgaris* that could be interesting for further research on octopus' nociception via an integrated computational pipeline. The authors succeed in laying out how they narrowed down their search to reach the final gene candidate. The authors are thorough in explaining their criteria and mapping the genes onto the octopus transcriptome.

The functional follow-up experiments performed are appropriate to support the author's claim that some of these genes that have orthologs in *C. elegans* are involved in acid-evoked nociception in the nematode. The authors use a drop assay to test turning behavior of individual nematodes to assess how mutations of certain proteins change their aversion to the acidic stimuli.

- Since pH is only one cause/activator of nociceptors, it would be intriguing and elevating the study to use a different assay in addition to test other forms of nociception (chemical, thermal, mechanical) for their further investigation in the octopus. However, the study succeeds in presenting the tested *O. vulgaris* ortholog and their involvement in nociception even without expanding into other regimes of nociception.

We are pleased that the Reviewer is of the view that the limited testing with a carefully controlled genotype/wildtype and aversive non aversive cue allows conclusion about an involvement in nociception. As raised by the fellow Reviewer wider testing would expand this information bases.

Indeed, we test a larger set of compounds including high osmolarity representative cues (4M fructose), volatile repellents (30% 1-octanol) and 30 mM CuSO₄ and these data are now shared in extended supplementary (Figs. S3-S4) to the manuscript. This is supported by modifications to the main text (introduction- line 83; methods - lines 200-203, 212-220; results - lines 355-374, discussion - lines 396-400, 404-406, 414, 454-455, 460-462).

However, the inclusion of only low pH in the original manuscript was based on two main reasons.

- 1) low pH is one of the few nociceptive cues already known to trigger a response in cephalopods and therefore relevant in their environmental context
- 2) Addressing a response to each of the cue we believed would have subtracted from the main message of showing how *C. elegans* can be a valuable platform to narrow down the search for relevant genes that should merit further functional characterisation (including follow up analysis of their role in more than a chemosensory response or modality)

- c) Were the data analyzed using appropriate statistical tests?

My main comment on the revision of this paper is the data in Figure 2. While I am convinced of the difference in aversion behavior of different mutant strains and their involvement in nociception, it's not fully clear to me why the behavior was scored as 0/1, since it could also be analyzed and presented as the number of reversals in the 5 s post-exposure (just as in figure 3). If the authors want to continue with the scoring system, I suggest the authors reconsider the data presentation and statistical analysis.

The reason for the adoption of a binary score was to reduce the potential background noise that could have been caused by the variability in the number of reversals among worms within the same strain and therefore was used as a more robust measure of responsiveness vs impaired responsiveness.

The use of a binary score or avoidance index (with responses scored between 0-1) has also been classically adopted by the authors who first established the test (see Hilliard, 2002; Hilliard 2004) and is still widely used as the standard method.

- In Figure 2, the statistical test used by the authors is a parametric ANOVA. Since the variable is binary (0,1) rather than continuous, a parametric ANOVA may not be optimal. The analysis here rather compares a probability that a nematode strain is highly responsive/avoidant to the stimulus (score 1, many reversals) vs. not highly responsive (score 0). So I think a binomial GLM (logistic regression) with a Benjamini-Hochberg or Holm-Bonferroni correction would be more appropriate, since the question is whether a mutation changes the probability of exhibiting frequent reversals, rather than comparing the averages. The wild type probability should be your reference in the model and you can check how likely it is that mutations change the probability for the worms to reverse frequently.

We thank the Reviewer for the suggestions. We have now redone the statistical analysis using a binomial GLM (logistic regression) with a Benjamini-Hochberg correction (Q:1%) and modified figures (Fig.1, Fig. S3 heatmap) and text (methods - lines 207-209) accordingly.

- As for the representation in Figure 2, the continuous scale on the y axis is misleading, since there is no other value than 0 or 1 in this analysis. So rather than reporting the means and SD which go beyond 1, I would suggest reporting the probability of being a frequent turner (score 1) based on the binomial model and indicate +/- 95% confidence interval around it. This should make the values not exceed 0 and 1.

As for the representation in Figure 2, we have again followed the advice suggesting to report the probability of being a responder based on the binomial model and the 95% CI around the mean.

- As for the representation in Suppl. Fig2, the representation of the binary with the violin plots does also not seem like the best way to present the data to me, since it is discrete and binary and not a real distribution that needs to be smoothed out by a violin plot. It suggests there is data between that does not exist. An alternative visualization—such as a

swarmplot, stripplot, or stacked barplot—might convey the discrete nature of the data more effectively.

We thank the Reviewer for the useful suggestion. We have modified the **figure S2** accordingly using a scatter dot plot rather than a violin plot which would have given the false impression of continuous data distribution.

We would like to clarify that even though Figure 2 and S2 were referring to the same experimental data, Figure S2 has been added following a request from the journal of representing individual data points.

Second decision letter

MS ID#: bio.062268R1

MS Title: Identification of molecular nociceptors in *Octopus vulgaris* through functional characterisation in *Caenorhabditis elegans*

Authors: Eleonora Maria Pieroni, Vincent O'Connor, Lindy Holden-Dye, Pamela Imperadore, Graziano Fiorito, James Dillon

I am happy to tell you that your manuscript has been accepted for publication in Biology Open, pending our standard publication integrity checks. It was accepted on 16th December 2025.